# Two conserved vocal central pattern generators broadly tuned for fast and slow rates generate species-specific vocalizations in *Xenopus* clawed frogs

**Ayako Yamaguchi\*, Manon Peltier**

School of Biological Science, University of Utah, Salt Lake City, United States

**Abstract** Across phyla, males often produce species-specific vocalizations to attract females. Although understanding the neural mechanisms underlying behavior has been challenging in vertebrates, we previously identified two anatomically distinct central pattern generators (CPGs) that drive the fast and slow clicks of male *Xenopus laevis,* using an ex vivo preparation that produces fictive vocalizations. Here, we extended this approach to four additional species, *X. amieti, X. cliivi, X. petersii, and X. tropicalis,* by developing ex vivo brain preparation from which fictive vocalizations are elicited in response to a chemical or electrical stimulus. We found that even though the courtship calls are species-specific, the CPGs used to generate clicks are conserved across species. The fast CPGs, which critically rely on reciprocal connections between the parabrachial nucleus and the nucleus ambiguus, are conserved among fast-click species, and slow CPGs are shared among slow-click species. In addition, our results suggest that testosterone plays a role in organizing fast CPGs in fast-click species, but not in slow-click species. Moreover, fast CPGs are not inherited by all species but monopolized by fast-click species. The results suggest that species-specific calls of the genus *Xenopus* have evolved by utilizing conserved slow and/or fast CPGs inherited by each species.

**\*For correspondence:**
a.yamaguchi@utah.edu

**Competing interest:** The authors declare that no competing interests exist.

## Editor's evaluation

This important paper compares the neural basis for different calling songs in five species of clawed *Xenopus* frogs using neural activity recordings combined with lesions of pathways and stimulation of specific parts of the circuit. The evidence supporting the claims is solid and reveals conservation and variation in the circuits generating fast and slow clicks in courtship calls. The work will be of broad interest to neurophysiologists beyond the vocalization topic.

## Introduction

Closely related species often exhibit dramatically different courtship behavior despite other similarities. This diversity of courtship behavior is a result of sexual selection, which leads to reproductive isolation and speciation (*Boughman, 2002*; *Ritchie, 2007*; *Ryan, 2021*). The nervous system of animals underlying courtship behavior is made up of both ancestral traits inherited through evolutionary lineage and derived traits selected to serve unique functions for a species. Identification of these traits can provide insights into the evolutionary trajectory underlying speciation. For instance, in crickets, the conserved and derived components of the courtship song neural circuitry are distributed along the abdominal ganglia, while in *Drosophila*, the conserved components are the command neurons that initiate the courtship song, and the derived components are the downstream thoracic neural networks (*Ding et al., 2019*). However, the strategies employed by the nervous system to

introduce behavioral diversity in vertebrates are not well understood, partly due to the lack of reduced preparation for detailed electrophysiological analyses.

The courtship vocalizations of the genus *Xenopus* offer a rare opportunity to explore the evolution of neural circuitry in vertebrates, as reduced preparations are available. All species of *Xenopus* produce species- and sex-specific vocalizations consisting of a series of clicks produced by the larynx specialized to generate sound under water (*Yager, 1992*; *Kwong-Brown et al., 2019*). Male *Xenopus* use advertisement calls containing a species-specific rate of clicks (0.6–150 Hz) (*Tobias et al., 2011*; *Evans et al., 2015*) to attract females, many of which are monophasic (i.e., clicks are repeated at a monotonous rate), while a few are biphasic (i.e. trains of clicks with two distinct rates are contained in a call). In contrast, female *Xenopus* produce 'release calls' consisting of slow clicks (<20 Hz) to escape from clasping males when not gravid. However, administering testosterone to adult female *X. laevis* results in the production of male-specific advertisement calls in 1–3 months (*Potter et al., 2005*).

Previously, we developed an ex vivo, isolated brain preparation from which fictive vocalizations can be elicited in male and female African clawed frogs, *X. laevis* (*Rhodes et al., 2007*). Vocalizations of *X. laevis* are generated by central pattern generators (CPGs; *Marder and Bucher, 2001*; *Rhodes et al., 2007*), neural networks that autonomously produce rhythmic motor programs in the absence of rhythmic central or sensory input (for reviews, see *Marder and Bucher, 2001*). The advertisement calls of male X. laevis is biphasic and are made of fast and slow trills containing clicks repeated at 60 and 30 Hz, respectively. The male *X. laevis* vocal pathways consist of the premotor nuclei in the parabrachial nucleus (PBN, formerly known as dorsal tegmental area of the medulla, DTAM) and the laryngeal motor nuclei, the nucleus ambiguus (NA), with extensive reciprocal connections (*Brahic and Kelley, 2003*). The advertisement call of male *X. laevis* consists of fast and slow trills containing clicks repeated at 60 and 30 Hz, respectively. We discovered that the fast and slow trills are generated by a pair of anatomically distinct CPGs contained in left and right hemi-brain in male *X. laevis*: fast trill CPGs contain neurons in the PBNs and NAs, whereas the slow trill CPGs are contained in the caudal brainstem including NAs (*Yamaguchi et al., 2017*).

The aim of the study was to determine if fast and slow trill CPGs discovered in male *X. laevis* are unique to this species or conserved across species of *Xenopus*. To this end, we developed ex vivo preparations in males of three additional species, *X. amieti, X. cliivi*, and *X. tropicalis*. Using these preparations, we examined the electrophysiological activity of the vocal neural circuitry during fictive advertisement calling in males of four species, including *X. petersii* (for which a fictive preparation had been previously developed; *Barkan et al., 2018*), which produce calls containing clicks repeated at variable rates. We also investigated if female *X. laevis* acquire fast trill-like CPGs or modify an existing CPG network in response to testosterone. Furthermore, we explored whether fast trill-like CPGs are present but remain latent as an evolutionary vestige in species that only produce slow clicks by examining the synapses that serve the critical function of the fast trill-like CPGs. We found that the two CPGs with conserved function and location are shared among species to generate species-specific courtship fast or slow clicks. Additionally, we found that fast trill-like CPGs are present only in species that produce fast clicks and their presence appears to be regulated by testosterone in these species.

## Results

### Isolated male brains of all species generate fictive advertisement calls resembling in vivo calls when exposed to appropriate stimuli

In this study, we used five species of *Xenopus* including *X. laevis*. Advertisement calls from the males of all five species are shown in *Figure 1A* as in vivo calls. The click rates and the number of clicks contained in advertisement calls of each species are summarized in *Figure 1B and C* as in vivo data. Out of the males of five species we studied, two species produced advertisement calls that contained clicks repeated only at rates >50 Hz (i.e. monophasic calls as previously described) – *X. amieti* (mean ± s.e., 143.0±2.90 Hz), *X. cliivi* (59.82±6.50 Hz), one species produced advertisement calls containing clicks repeated only at rates <35 Hz – *X. tropicalis* (31.9+1.18 Hz), and two species produced advertisement calls with both fast (>50 Hz) and slow (<35 Hz) clicks (i.e. biphasic calls as previously described) – *X. petersii* (69.9+2.02 Hz, 31.3+1.99 Hz), and *X. laevis* (58.3+2.47 Hz), 38.4+2.98 Hz.

Next, we developed a method to obtain fictive vocalizations from isolated brains of the males of all the *Xenopus* species used in this study. Previous studies had shown that application of serotonin

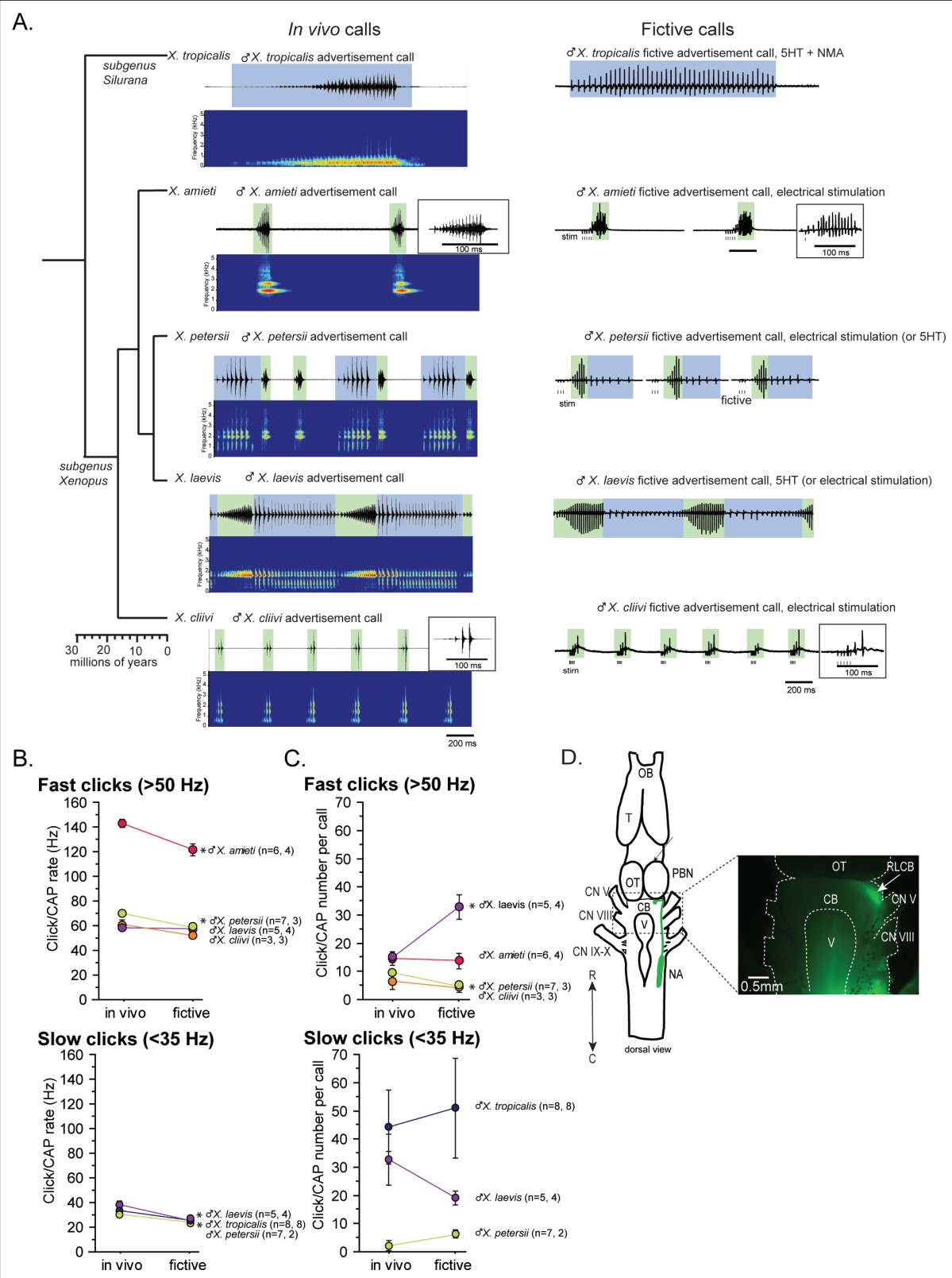

**Figure 1.** Phylogeny of five species of *Xenopus* and their advertisement calls. (**A**). Chronogram, in vivo vocalizations, and fictive vocalizations recorded from males of each species. Left column: chronogram based on mitochondrial DNA (Modified from *Evans et al., 2015*) of studied species. Middle column: amplitude envelope (top panel) and sound spectrogram (bottom panel) of in vivo advertisement calls. Right column: fictive advertisement calls recorded ex vivo from the isolated brain. The green and blue background of traces indicate fast (>50 Hz) and slow (<35 Hz) clicks, respectively,

*Figure 1 continued on next page*

*Figure 1 continued*

and the vertical lines below the fictive vocalizations labeled with 'stim' indicate electrical stimuli applied to the rostral-lateral cerebellum (RLCB, **D**) to elicit fictive calls in some brains. The type of stimulus following the species name indicates what was used to elicit the fictive calls shown, and those in parenthesis are other stimulus types that are also effective in eliciting fictive calls. (**B**). Comparison of click and compound action potential (CAP) rates of calls recorded in vivo and ex vivo from each species. Each circle with an error bar indicates the mean + s.e. of click or CAP rates (in Hz). For both **B** and **C**, asterisks preceding the species name indicate significant differences between the rates of clicks/CAPs produced in vivo and ex vivo. Note that some error bars are hidden behind the data points due to the small size. (**C**). The number of clicks/CAPS in a call recorded in vivo and ex vivo. Each circle with an error bar indicates the mean + s.e. of click/CAP number. The first and the second sample sizes after species name refers to in vivo and fictive data, respectively. (**D**). The location of the rostral-lateral cerebellum (RLCB), a site that is effective in eliciting fictive calls when stimulated electrically. Left; a cartoon showing the dorsal view of an isolated brain of *Xenopus*. The double headed arrow indicates rostral (R) and caudal (C) orientation of the brain. When dextran dye was injected into the nucleus ambiguus (NA), axons of projection neurons in the NA and the parabrachial nucleus (PBN) that project reciprocally to each other are labeled and can be viewed from the dorsal surface of the brain as seen in the photo on the right. The area at the lateral edge of the cerebellum (CB) along the labeled projections is the RLCB (white arrow). Delivering stimulus pulses to this area using a concentric electrode elicits fictive calls in most brains except in the brain of *X. tropicalis*. CB; cerebellum, CN V; cranial nerve V, NA; nucleus ambiguus, OB; olfactory bulb. OT; optic tectum, PB; parabrachial nucleus, T; telencephalon.

The online version of this article includes the following figure supplement(s) for figure 1:

**Figure supplement 1.** Types of stimuli used to elicit fictive advertisement calls.

(5HT) to the isolated brains of male *X. laevis* and *X. petersii* readily elicited fictive advertisement calls (*Rhodes et al., 2007*; *Barkan and Zornik, 2019*). In this study, we found that we could also induce fictive advertisement calls from the isolated brains of three additional species (*Figure 1A*, right column), but the types of stimuli required varied depending on species. For instance, 5HT alone did not elicit fictive calls from the brains of *X. tropicalis* (*Figure 1—figure supplement 1A*, n=7), but a combination of 5HT and N-methylaspartate (NMA, *Figure 1A*, right column, *Figure 1—figure supplement 1A*) was effective. In contrast, neither 5HT alone nor 5HT and NMA together induced fictive calls from the isolated brains of *X. amieti* and *X. cliivi* (*Figure 1—figure supplement 1B*, top trace, for example). However, trains of electrical pulses delivered to the rostral-lateral cerebellum (RLCB, *Figure 1D*) readily evoked fictive advertisement calls (*Figure 1A*, right column, *Figure 1—figure supplement 1B*). The RLCB is the region in the brain where the ascending and descending axons of the projection neurons in the parabrachial nucleus (PB) and the nucleus ambiguus (NA) are located close to the surface of the brain. Dextran dye injected into NA that labeled anterograde and retrograde axons allowed us to visualize this location (*Figure 1D*). Electrically stimulating RLCB likely activates a previously unidentified component of the central vocal pathways that initiates vocalizations. Interestingly, electrical stimulation delivered to the RLCB was effective in evoking fictive advertisement calls from all males tested in this study (*Figure 1—figure supplement 1C*, for example), except for male *X. tropicalis* (*Figure 1—figure supplement 1A*, n=6). The stimulus pulse duration used was 40us with a frequency ranging from 10 to 100 Hz, amplitudes varying from 0.7 to 6mA, and a pulse count ranging from 3 to 30. Effective stimulus parameters varied greatly between preparations, and there was no observable difference in the effective parameters between species.

The temporal structure of fictive advertisement calls closely resembled the calls recorded in vivo, as shown by comparing the sound amplitude envelope (top traces for each species) on the left column and the fictive call traces on the right column in *Figure 1A*. Although the exact rate of compound action potentials (CAPs) is significantly slower (*Figure 1B*) and the number of CAPs contained in fictive calls differ from the calls recorded in vivo in some species (*Figure 1C*), the overall temporal structure is well preserved in all species.

## Parabrachial nuclei activity coincides with fictive advertisement calls that contain compound action potentials repeated at frequency above 50Hz

Previously, we found that in male *X. laevis*, the parabrachial nucleus (PBN) is active during fictive fast trills, but remains inactive or exhibits minimal activity (less than twice the amplitude of the noise) during fictive slow trills. (*Yamaguchi et al., 2017*; *Figure 2Aiv*). Here, we investigated whether this observation applies to males of other species including *X. amieti*, *X. cliivi*, *X. petersii*, and *X. tropicalis*. The results showed that in all these species, the PBN was predominantly active during fictive calls that contain compound action potentials (CAPs) repeated at rates faster than 50 Hz (indicated by traces

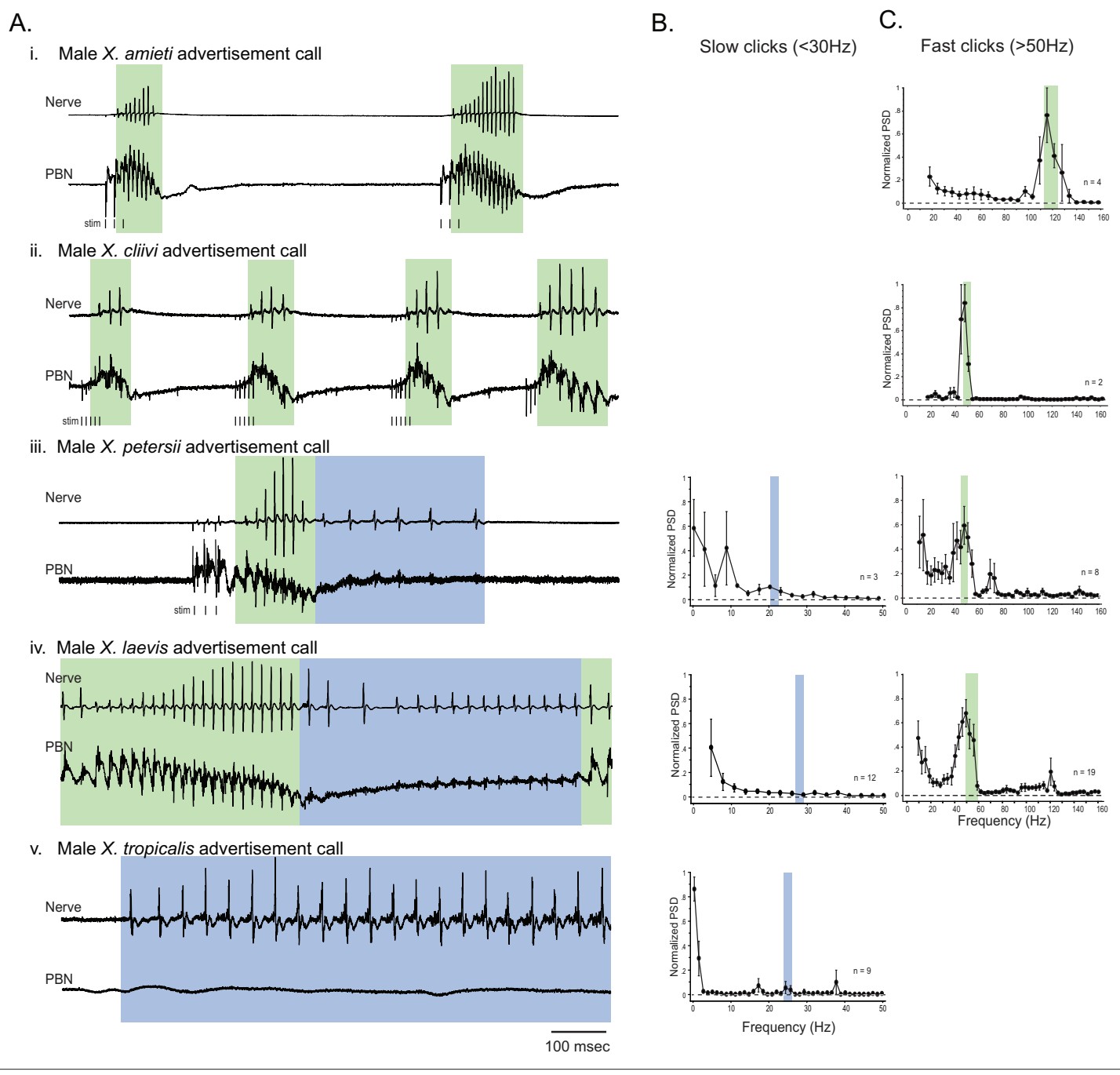

**Figure 2.** The activity of the parabrachial nucleus during fictive advertisement calls. (**A**). Local field potential (LFP) recordings obtained from the parabrachial nucleus (PBN) during fictive advertisement calling in male *X. amieti* (i), male *X. cliivi* (ii), male *X. peterii* (iii), male *X. laevis* (iv), and male *X. tropicalis* (v). Top traces; laryngeal nerve recording, bottom traces: PBN LFP recordings. The green and blue backgrounds indicate fast (>50 Hz) and slow (<35 Hz) compound action potentials (CAPs), respectively. Vertical lines below the traces labeled with 'stim' indicate the timing of electrical pulses delivered to the RLCB (*Figure 1D*) to elicit fictive advertisement calls in some cases. (**B**). mean power spectral density (PSD) of PBN LFP recordings during fictive slow clicks seen in **A**. Blue frames show the mean ± std of the CAP rates for slow clicks. (**C**). Mean PSD of PBN LFP recordings during fictive fast clicks seen in **A**. Green frames show the mean ± std of the CAP rates for fast clicks.

with a green background in *Figure 2A*), but not during fictive calls containing CAPs repeated at rates below 35 Hz (indicated by traces with blue background in *Figure 2A*). In male *X. amieti*, *X. cliivi*, and *X. petersii* a PBN local field potential (LFP) showed phasic activity that correlated with the CAPs during fictive calls containing CAPs faster than 50 Hz, similar to what was observed in male *X. laevis* during

fictive fast trills (indicated by all traces with green background in *Figure 2A*). The mean power spectral density (PSD) of the PBN LFP waveform (normalized to the maximum power for each animal) showed a clear peak at the frequency corresponding to the CAP repetition rates, as was also observed in male *X. laevis* (*Figure 2C*). However, in specie with slow clicks in their advertisement calls (slow trills of male *X. petersii* and the advertisement calls of male *X. tropicalis*), the PBN activity was mostly absent during fictive slow clicks (*Figure 2A*, traces with blue background). In a few brains, some PBN LFP phase-locked to CAPs was observed during slow CAPs (*Figure 2A*, see LFP during slow trills of male *X. petersii* and male *X. laevis*), but the LFP amplitude was significantly lower than that during fictive fast clicks (*Figure 2A*, see the amplitude of LFP during slow trills of male *X. petersii* and male *X. laevis* compared to that during fast trills). Consequently, the mean normalized PSD of the PBN LFP recorded during fictive slow clicks showed no clear peak (*Figure 2B*).

For brevity, we will refer to all clicks and compound action potentials (CAPs) repeated at a rate ≥50 Hz as 'fast clicks/CAPs', and the species that generate them (male *X. amieti*, *X. cliivi*, *X. petersii*, and *X. laevis*) as 'fast clicker' even if their advertisement calls also include slow clicks (i.e. male *X. petersii* and male *X. laevis*). Similarly, we refer to all clicks and CAPs repeated at a rate ≤35 Hz as 'slow clicks/CAPs' and the species that produce only slow clicks (i.e. male *X. tropicalis*) as 'slow clickers'.

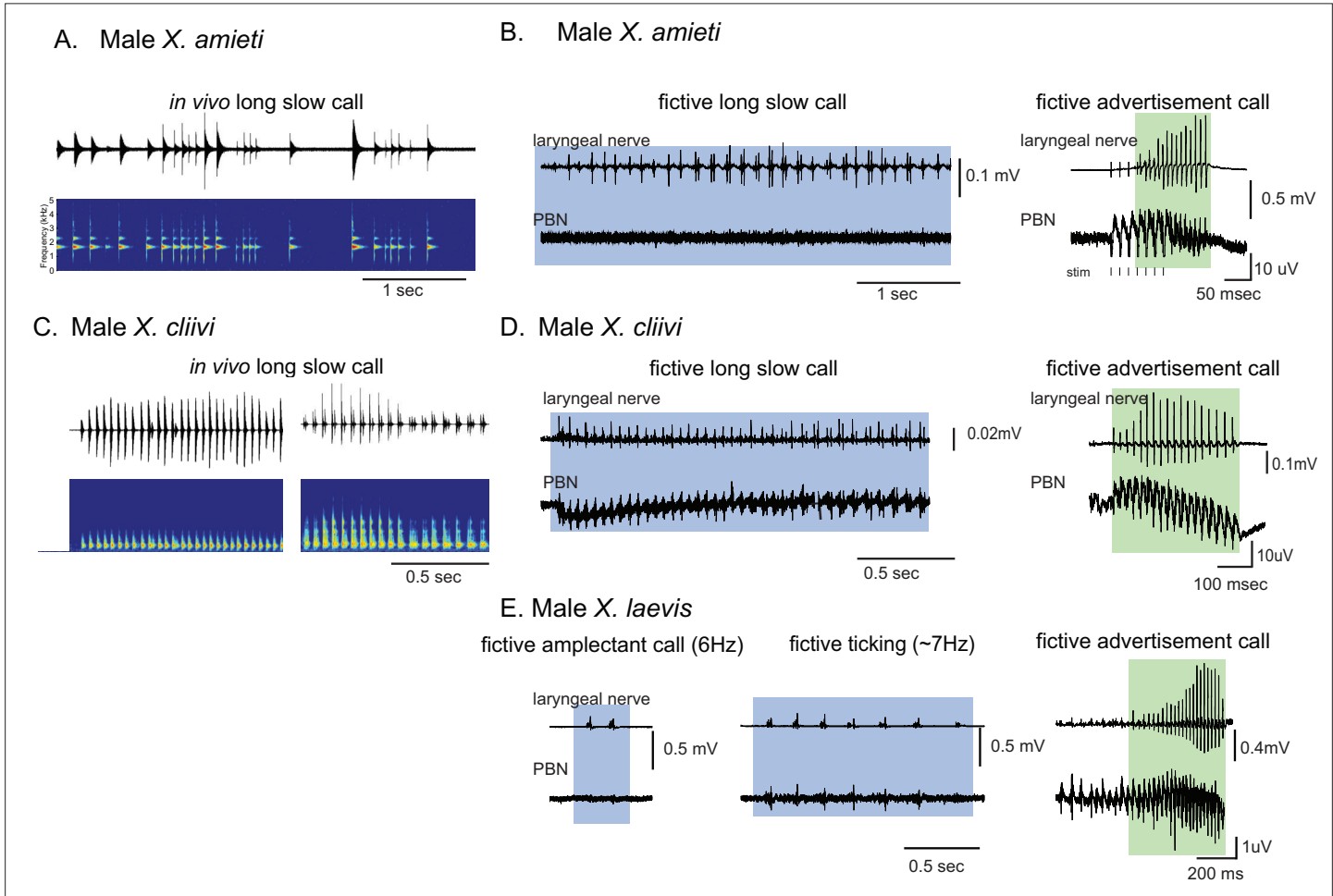

**Figure 3.** Calls produced by fast clickers containing slow (<35 Hz) clicks are not accompanied by salient parabrachial nucleus activity. (**A, C**), Amplitude envelope (top) and the sound spectrogram (bottom) of a 'long slow call' produced by male *X. amieti* (**A**) and male *X. cliivi* (**C**) in vivo. (**B, D**), Presumed fictive long slow call(left with blue background) and fictive advertisement call (right with green background) obtained from the same brain of a male *X. amieti* (**B**) and male *X. cliivi* (**D**). Top; Laryngeal motor nerve recordings, bottom; parabrachial nucleus (PBN) local field potential (LFP) recordings. The same Y scale for the PBN LFP (but with an extended X scale) is used for both fictive calls for ease of amplitude comparison. (**E**). A fictive amplectant call (left), fictive ticking (middle), and fictive advertisement call (right) recorded from a brain of the same male *X. laevis*. Top; laryngeal nerve recording, bottom; PBN LFP recording. The Y scale for the PB LFP recordings (but not the X scale) is the same for all three recordings for ease of amplitude comparison.

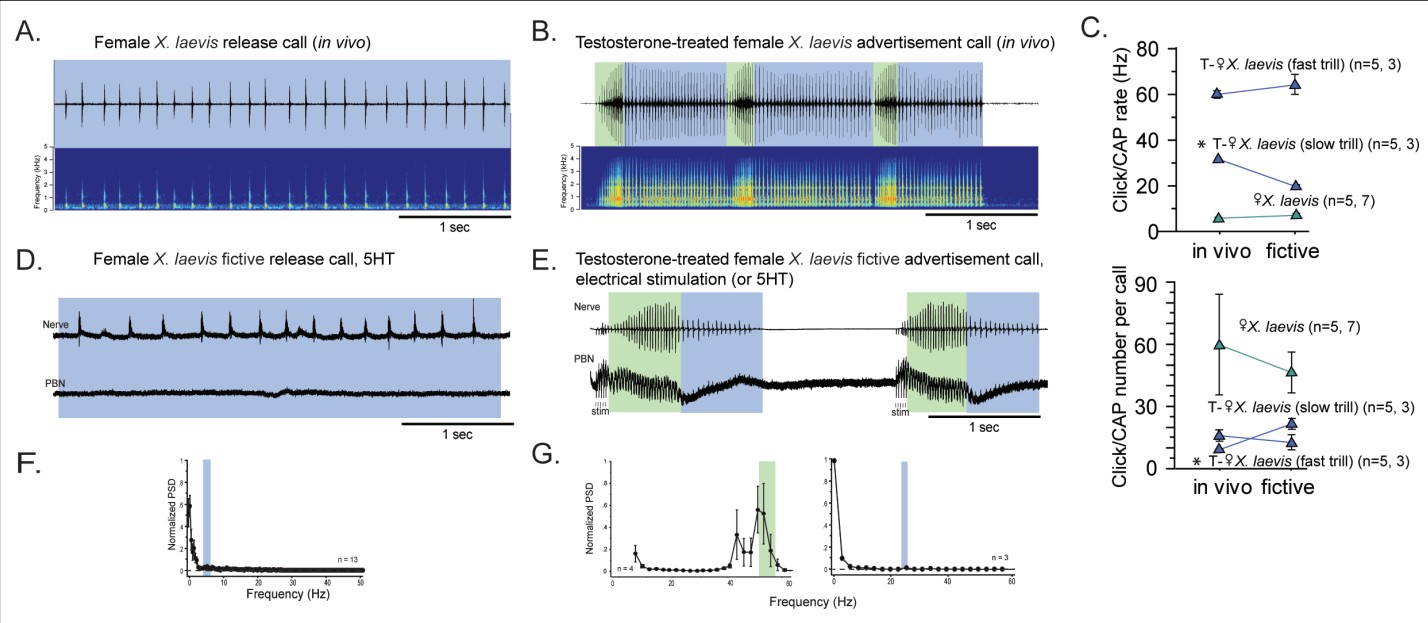

**Figure 4.** The parabrachial nucleus that is silent during release calling in female *X. laevis* becomes active after testosterone-induced vocal masculinization. (**A, B**) Amplitude envelope (top) and sound spectrogram (bottom) of a release call (**A**) and advertisement call (**B**) recorded in vivo from female *X. laevis* and testosterone-treated female *X. laevis*. The blue and green background behind the amplitude envelope indicates slow (<35 Hz) and fast (>50 Hz) clicks, respectively. (**C**) Click repetition rates and click number per call recorded in vivo and ex vivo from female *X. laevis* and testosterone-treated female *X. laevis*. Each triangle with an error bar indicates the mean ± s.e. of click repetition rate (top) and click number per call (bottom). The names of the animal preceded by an asterisk indicate a significant difference between the in vivo and fictive calls. The sample sizes following species name refer to in vivo and fictive data, respectively. (**D, E**) Fictive release call and advertisement calls recorded from an isolated brain of a female *X. laevis* and testosterone-treated female *X. laevis*. Top trace; laryngeal nerve recordings, bottom trace; local field potential (LFP) recordings obtained from the parabrachial nucleus (PBN). The blue and green background behind the amplitude envelope indicates slow (<35 Hz) and fast (>50 Hz) clicks, respectively. The type of stimulus following the species name indicates what was used to elicit the fictive calls shown, and that in parenthesis is another stimulus type that is also effective in eliciting fictive calls. Compare the PBN activity during fast trills to those recorded from the intact female during fictive release calling. (**F, G**) Mean power spectral density (PSD) of PBN LFP recordings during fictive release calling in female *X. laevis* (**F**) and fictive advertisement calling in testosterone-treated female *X. laevis* (**G**). Blue and green frames show the mean + std of the CAP rates for fictive release calls, slow trills, and fast trills.

It is important to note that the vocal repertoire of *Xenopus* species includes calls other than advertisement calls. Male *X. laevis*, for example, produce amplectant clicks (10 Hz) when clasping gravid females (*Tobias et al., 2004*) and 'ticking' when clasped by a male (*Tobias et al., 2014*). In this study, we discovered that male *X. cliivi* and *X. amieti* produce novel calls containing clicks repeated at 6–20 Hz (with single or double clicks as repetition units) in the presence of conspecific males. We named these calls 'long-slow calls' (*Figure 3A and C*). Fortuitously, we recorded these fictive calls containing slower CAP rates generated spontaneously from isolated brains, including long-slow calls from one male *X. amieti* (*Figure 3B*) and two male *X. cliivi* brains (*Figure 3D*), amplectant clicks from four male *X. laevis* (*Figure 3E* left), and ticking from three male *X. laevis* (*Figure 3E* middle). When PBN LFP recordings during these slow fictive calls were examined, we found that the PBNs were either silent (*Figure 3B and E* amplectant call) or showed activity (*Figure 3D and E*, release calls) significantly lower in amplitude than activity accompanying fictive fast clicks (*Figure 3B, D and E* right panels). These results suggest that, regardless of the vocal repertoire of the species, the PBN does not play a significant role, if any, in producing fictive slow clicks (<35 Hz) compared to its role in producing fast clicks (>50 Hz).

## After testosterone-induced vocal masculinization, the parabrachial nucleus of female *Xenopus laevis*, which was silent during fictive release calls, becomes active

Previously, we showed that adult female *X. laevis*, which normally produce release calls containing clicks trains at a rate of ~6 Hz (*Figure 4A*), can generate male-like advertisement calls within 1–3 months of testosterone treatment (*Figure 4B*; *Potter et al., 2005*). Our objective was to determine whether female *X. laevis* use slow trill-like central pattern generators (CPGs), like those observed in male *X. laevis*, to produce release calls. Additionally, we aimed to determine if females acquire fast trill-like CPGs or continue to use the existing slow trill-like CPGs to produce masculinized fast click calls. To this end, we examined the activity of parabrachial nucleus (PBN) during vocal production in control and testosterone-treated female *X. laevis*.

The application of serotonin (5HT) elicited fictive release calls and advertisement calls from the isolated brains of control and testosterone-treated female X. laevis, respectively (*Figure 4D and E*). In testosterone-treated females, electrical stimulation delivered to the rostral-lateral cerebellum (RLCB) also elicited fictive advertisement calls. Although the exact rate and the number of compound action potentials (CAPs) in the fictive calls are significantly different in some cases (*Figure 4C*), the overall temporal structure resemble those of calls recorded in vivo (*Figure 4A, B, D and E*). In control females, the PBN remained silent during fictive release calls in all animals tested (n=13, *Figure 4D* bottom trace). The mean power spectral density (PSD) of the PBN local field potential (LFP) during fictive release calls showed no peak (*Figure 4F*). In contrast, in testosterone-treated female *X. laevis*, the PBN showed phasic activity coinciding with CAPs during fast trills, but not during slow trills (*Figure 4E*), similar to male *X. laevis*. The mean PSD of the PBN LFP during fast trills had peaks between 50 and 60 Hz (*Figure 4G* left graph) while no peak is evident during slow trills (*Figure 4G* right graph), like male *X. laevis* (*Figure 2A and C*). Thus, we conclude that testosterone enables the recruitment of female PBNs to generate the phasic activity that accompanies fictive fast trills. Results obtained from males and females *Xenopus* show that the presence of PBN activity is associated with the CAP repetition rates. During fictive calls with CAP rates greater than 50 Hz, PBN shows activity phase-locked to the CAPs, whereas during fictive calls with CAP rates slower than 35 Hz, the PBN shows almost no activity in all species and sexes examined. Throughout the rest of our analyses, we refer to control and testosterone-treated female *X. laevis* as slow and fast clickers, respectively.

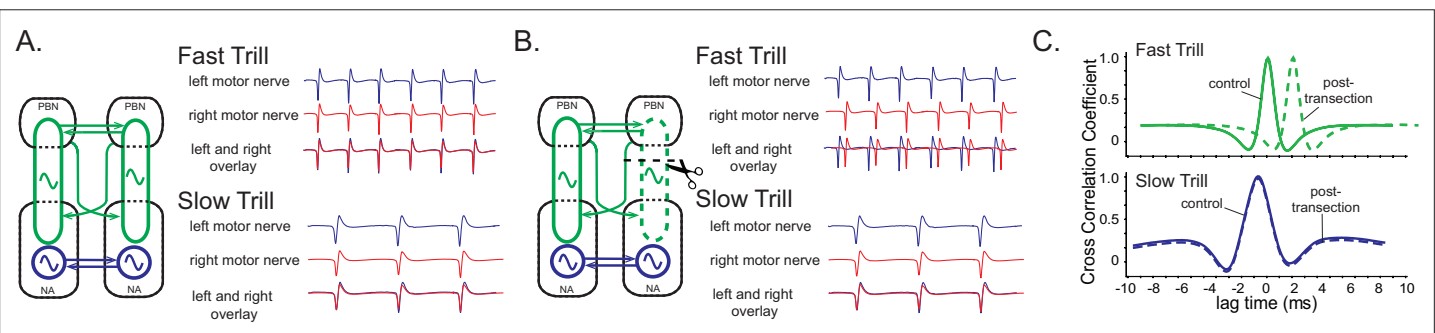

**Figure 5.** A model of fast trill and slow trill central pattern generators (CPGs) in male *Xenopus laevis* based on a previous study (*Yamaguchi et al., 2017*). (A). Left and right fast trill CPGs (green) span across the parabrachial nucleus (PBN) and the nucleus ambiguus (NA) whereas the slow trill CPGs (blue) are contained in the NA. The left and right CPGs coordinate their activity via reciprocal projection (shown as double-headed horizontal arrows). During fast and slow trills of a fictive advertisement call, compound action potentials (CAPs) produced by the left (blue) and right (red) laryngeal nerves are synchronous. (B). When a right PBN and right NA are transected (scissors with dotted line), the right fast trill CPGs become dysfunctional while the left fast trill CPGs and both slow trill CPGs remain functional. When a fictive advertisement call is elicited from the transected brain, the CAPs produced by the nerve on the transected side (right) were delayed compared to those produced by the nerve on the intact side (left) during fast trill, but not during slow trills. This delay is likely caused by the fact that the laryngeal motoneurons on the transected side (right) are driven by the fast trill CPGs on the intact (left) side during the fast trill. However, during slow trill, slow trill CPGs on both sides remain functional even after the transection, and therefore, there is no delay between the two nerves. (C). Cross-correlation coefficient as a function of lag time between left and right laryngeal nerve recording. In the control brain, the maximum cross-correlation coefficient is zero, and the activity of the two nerves is synchronous. The lag time of the maximum cross-correlation coefficient becomes positive during fast, but not during slow trills after the transection.

## Unilateral transection between the parabrachial nucleus and the nucleus ambiguus desynchronizes left and right laryngeal nerve activity during fast, but not slow clicks in five *Xenopus* species

Previously, we showed that transected left and right hemi-brains of male *X. laevis* can generate both fast and slow trills, indicating that there are pairs of fast and slow trill central pattern generators (CPGs) in the left and right brainstem (*Figure 5A*, left schematic; *Yamaguchi et al., 2017*). When we unilaterally transected the projections between the parabrachial nucleus (PBN) and the nucleus ambiguus (NA) (*Figure 5B*, left schematic), the brain still produced fictive advertisement calls, but the compound action potentials (CAPs) from the two nerves desynchronize during fast trill only, not during slow trills (*Figure 5B and C Yamaguchi et al., 2017*). Based on these results, we concluded that the fast trill CPGs span between the PBN and NA (*Figure 5A* left, green oscillators), whereas the slow trill CPGs are confined to the caudal brainstem (*Figure 5A* left, blue oscillators). When fast trill CPGs on the transected side become dysfunctional, signals from the functional fast trill CPGs on the intact side are transmitted to the non-functional side during fast trills, causing a delay in the production of the CAPs on the transected side compared to the intact side (*Figure 5B and C*; *Yamaguchi et al., 2017*). However, CAPs during slow trills of the transected brain remain unaffected since slow trill CPGs on both sides are functional. Here, we investigated the effect of unilateral transection between the PBN and NA on CAP synchrony in fast and slow clickers to determine if fast trill-like and slow trill-like CPGs are present in the brains of fast and slow clickers.

First, we confirmed that CAPs recorded from the left and right laryngeal nerves are synchronous during fictive calls (a measure of CAP synchrony, see Methods). When the maximum lag time between the left and right CAPs (a measure of synchrony, see Materials and methods) during both fictive slow and fast clicks were compared, they did not differ significantly from zero (slow clicks: one-sample sign test, p=0.557, n=26 including 11 male *X. tropicalis*, 4 male X. petersii, 9 female *X. laevis*, and 2 testosterone-treated female *X. laevis Figure 6G* control, fast clicks: P>0.790, n=14 including 4 male *X. amieti*, 3 male *X. cliivi*, 5 male *X. petersii*, and 2 testosterone-treated female *X. laevis, Figure 7E* control), indicating that CAPs recorded from the two nerves are synchronous during both fast and slow clicks in all intact brains. We also verified the completeness of the unilateral transection anatomically (*Figure 6A*) by depositing fluorescent dextran into the NA post-hoc and checking the absence of the labeled soma and axon terminals in the PBN on the transected side (*Figure 6B*).

During fictive slow clicks (male *X. tropicalis* advertisement call, male *X. petersii* slow clicks, testosterone-treated female *X. laevis* slow clicks, and female *X. laevis* release calls), the unilateral transection did not significantly change the mean maximum lag time between the CAPs recorded from the two nerves (*Figure 6*, Wilcoxon signed rank test, Z=–0.417, p=0.677, n=25 including 11 male *X. tropicalis*, 3 male *X. petersii*, 9 female *X. laevis*, 2 testosterone-treated female *X. laevis*), suggesting that the projections between the PBN and the ipsilateral NA are not necessary for the function of the slow click CPGs.

During fast clicks, in contrast, the CAPs became desynchronized after the transection, with the CAPs recorded from the nerve on the transected side lagging behind those recorded from the intact side (*Figure 7A* - D). The mean maximum lag time between the two nerves increased significantly after the transection (*Figure 7E*, Wilcoxon signed rank test, Z=–2.93, p=0.003, n=11 including 3 male *X. amieti*, 2 male *X. cliivi*, fast clicks of 4 male *X. petersii,* and fast clicks of 2 testosterone-treated female *X. laevis*), with the mean lag between the CAPs on the intact and transected side increased by 1.53+0.344 ms (mean ± s.e., n=11) after the transection. These results support the notion that fast clicks are mediated by central pattern generators (CPGs) that span between PBN and NA, whereas slow clicks are generated by CPGs located caudal to the transection. In other words, anatomically distinct fast and slow trill-like CPGs found in male *X. laevis* appears to be conserved across species to generate fast and slow clicks, respectively.

## Fast trill-like central pattern generators on the intact side of the brain likely drive laryngeal motoneurons on the transected side during fast clicks

To determine if the lag between the compound action potentials (CAPs) recorded on the two nerves after the transection is due to the time it takes for signal from the parabrachial nucleus (PBN) on the

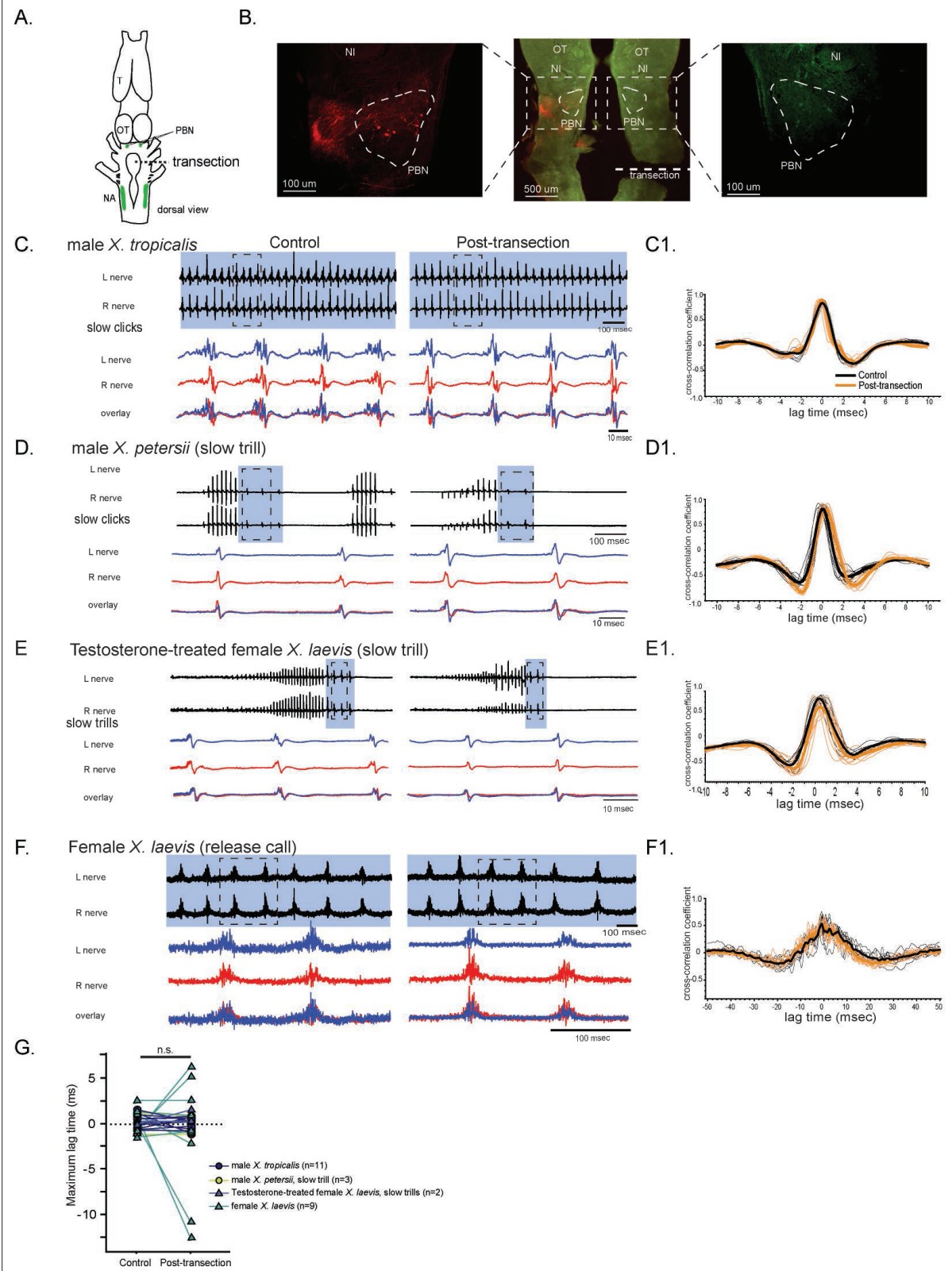

**Figure 6.** The unilateral transection between the parabrachial nucleus (PBN) and the nucleus ambiguus (NA) did not change the synchrony of the compound action potentials (CAPs) during fictive slow clicks. (**A**). A cartoon showing the dorsal view of an isolated *Xenopus* brain, showing the location of the transection that was made between the PBN and NA on the right side with dotted black line. (**B**). Histological sections of the transected brain. After the electrophysiological experiments, Texas red dextran and fluorescein dextran were deposited into the left and right NA, respectively, to verify

*Figure 6 continued on next page*

*Figure 6 continued*

that the transection had been completed successfully. The middle panel shows the low-magnification view of the horizontal section of the rostral brainstem showing PBNs on both sides in dotted triangles. Left and right images show magnified view of the PBN with labeled soma and axons on the intact side on the left, and no labeled cells or processes on the transected side on the right. (**C - F**). Fictive advertisement calls produced by male *X. tropicalis* (**C**), male *X. petersii*(**D**), testosterone-treated female *X. laevis* (**E**), and fictive release calls produced by female *X. laevis* (**F**) before (left column) and after (right column) the unilateral transection. Top two traces; left and right laryngeal nerve recordings. The blue background indicates slow CAPs. Areas in dotted rectangles are enlarged in the bottom three traces, the left nerve in blue, the right nerve in red, and both traces are overlaid in the bottom. C1 – F1. Cross-correlation coefficients as a function of the lag between the intact and the transected nerves during fictive advertisement calls produced by male *X. tropicalis* (**C1**), slow clicks of advertisement call produced by male *X. petersii*(**D1**), testosterone-treated female *X. laevis* (**E1**), and fictive release calls produced by female *X. laevis* (**F1**). Thick lines indicate the mean of control (black) and post-transection (orange) conditions, and thin lines indicate individual data. Note that the timing of the maximum crosscorrelation coefficient is centered around zero both before and after the unilateral transection. (**G**). Mean maximum lag time after the transection during fictive slow clicks produced by male *X. tropicalis* (n=11), male *X. petersii* (n=3), testosterone-treated female *X. laevis* (n=2), and female *X. laevis* (n=9) did not differ significantly from that before the transection.

intact side to cross the midline to contralateral motoneurons on the transected side, we stimulated the intact-side PBN before and after the transection and measure the latency between the stimulus onset and the peak of the CAPs recorded from the two nerves (*Figure 7F*). In intact brains of fast clickers, a train of stimulus (40usec pulses, 30 Hz) delivered to the PBN elicits CAPs from both nerves simultaneously with a latency of ~6.5 ms that did not differ between the two nerves (Wilcoxon signed rank test, $Z=–1.60$, $p=0.101$, n=9 including 1 male *X. laevis*, 3 male *X. amieti*, 2 male *X. cliivi*, 1 male *X. petersii*, and 2 testosterone-treated female *X. laevis*). After the unilateral transection, PBN stimulation on the intact side still elicited CAPs from both nerves. However, the amplitude of CAPs recorded on the transected side was significantly lower than before the transection, whereas no significant change was detected on the intact side (data not shown). In addition, the latency of the CAPs on the intact side remained unchanged after the transection (*Figure 7G*, left graph, Wilcoxon signed rank test, $Z=–0.178$, $p=0.859$, n=9 including 1 male *X. laevis*, 3 male *X. amieti*, 2 male *X. cliivi*, 1 male *X. petersii*, and 2 testosterone-treated female *X. laevis*). After the unilateral transection, PBN stimulation on the intact side still elicited CAPs from bo, but on the transected side, the transection significantly elongated the latency (*Figure 7G*, right graph, Wilcoxon signed rank test, $Z=–2.67$, $p=0.008$, n=9), with an average increase of $1.79±0.49$ ms (mean ± s.e.). Importantly, the increase in the PBN-evoked CAP latency on the transected side was not significantly different from the lag between the right and left CAPs during fictive fast clicks after the transection (*Figure 7H*, Mann-Whitney U test, $Z=–1.254$, $p=0.210$, n=20). These results show that the signals from the PBN on the intact side can reach the contralateral laryngeal motoneurons on the transected side, with a delay similar to those observed during fictive fast clicks. Therefore, the CAPs recorded from the nerve on the transected side during fictive fast clicks are likely driven by the fast trill-like CPGs on the intact side.

## All fast clickers share fast, potentiating monosynaptic connections between the parabrachial nucleus and the laryngeal motoneurons

Previously, it was shown that the synapses between the projection neurons in the parabrachial nucleus (PBN) and the laryngeal motoneurons in male *X. laevis* are glutamatergic and strong (*Zornik and Kelley, 2008*). Here, we determined if the synapses from the PBN to laryngeal motoneurons exhibit similar properties in all fast clickers.

To address this question, we first characterized the synapses between the PBN and the laryngeal motoneurons in male *X. laevis* by unilaterally stimulating the PBN while recordings are obtained from both laryngeal nerves. We found that in11 out of 13 male *X. laevis* brains, compound action potentials (CAPs) were recorded from both nerves (*Figure 8A*), while the remaining animal showed CAPs only from either contralateral or ipsilateral nerve. A stimulus train at a frequency below 10 Hz did not elicit any CAPs (*Figure 8D*, see 1 Hz stimulation), but at a frequency above 10 Hz, we observed a train of CAPs that progressively potentiated and later plateaued in amplitude (*Figure 8A, D and E*) while gradually diminishing and leveling off in latency (defined as the time between the stimulus onset and the CAP peak as indicated as a double arrow in *Figure 8D and E*). These changes in CAP amplitude and latency were successfully fitted with exponential curves (mean ± s.e. cross-correlation coefficient of the fit = $0.781 ± 0.029$ for CAP amplitude, $0.938+0.01$ for CAP latency, *Figure 8E*) with mean $\tau$ of $6.90±0.995$ pulses for CAP amplitude and $3.64±.515$ (mean ±s.e.) pulses for CAP

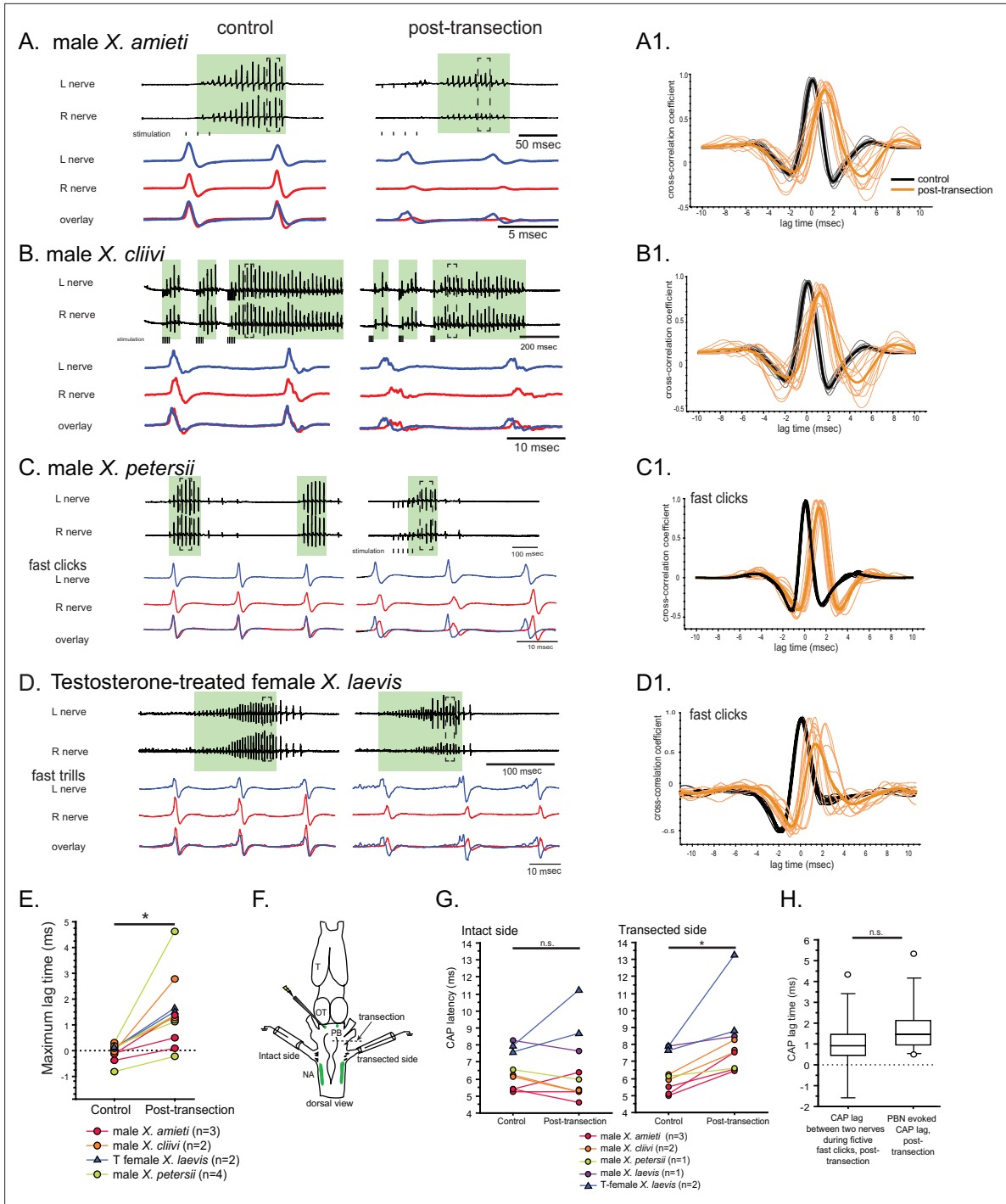

**Figure 7.** The unilateral transection between the parabrachial nucleus (PBN) and the nucleus ambiguus (NA) resulted in delayed compound action potentials (CAPs) during fictive fast clicks. (A – D), Fictive advertisement calls recorded from male *X. amieti* (A), male *X. cliivi* (B), male *X. petersii* (C), and testosterone-treated female *X. laevis* (D) before (left) and after (right) the unilateral transection. The top two traces are the laryngeal nerve recordings obtained from the left and right nerves. Green background indicates fictive fast clicks. Areas in dotted rectangles are enlarged in the third, fourth, and fifth traces. In these enlarged traces, the left nerve is shown in blue, the right nerve is shown in red, and both traces are overlayed at the bottom. A1 through D1 shows the cross-correlation coefficients as a function of the lag between the intact and the transected nerves during fictive advertisement calls produced by male *X. amieti* (A), male *X. cliivi* (B), fast trills of male *X. petersii* (C), and testosterone-treated female *X. laevis* (D) before (black lines) and after (orange lines) the transection. Thick lines indicate the mean and thin lines indicate individual data. (E). Mean maximum lag time for the fast clickers. After the transection, the maximum lag time significantly increased during fast clicks in the fast clickers. (F). A cartoon showing the dorsal view of the brain showing the stimulation of the intact parabrachial nucleus (PBN) before and after the transection was made between the right PBN and the

*Figure 7 continued on next page*

Figure 7 continued

right nucleus ambiguus (NA) while recordings are obtained from both nerves. (**G**). CAP latency, defined as the time between the stimulus onset and the time of the CAP peak, recorded from the nerve on the intact side and on the transected side before and after the transection. The latency was measured after the 15th stimulus pulse after it reached a plateau (See *Figure 8E*). The CAP latency on the intact side did not change, but those on the transected side increased significantly after the transection. (**H**). The box plot showing the increase in the CAP lag after the transection during fictive fast clicks (CAP lag during fictive fast clicks) and the PBN-evoked CAP latency increase after the transection (PBNevoked CAP lag). They did not differ significantly.

latency that was independent of stimulus frequencies (*Table 1*). In other words, when PBN was stimulated at stimulus frequency >10 Hz, 95% of the maximum CAP amplitude was reached within ~21 pulses and 95% of the latency was reached within 11 pulses (*Figure 8E*). These progressive changes in amplitude and latency likely represent the increasing recruitment of motoneurons and synaptic facilitation, respectively. Interestingly, the CAP latency recorded from the ipsilateral and contralateral nerves (measured in response to stimulus frequency >10 Hz after the 11th stimulus pulse) was similar (*Figure 8B*, Wilcoxon Signed Rank test, Z=–0.706, p=0.480, n=12), indicating the presence of bilateral projections from PBN to left and right laryngeal motoneurons driving synchronous firing of motoneurons on both sides (*Zornik and Kelley, 2007*). Importantly, the maximum amplitude of the CAPs evoked in response to PBN stimulation was comparable to those observed during fictive fast trills (*Figure 8C*, Wilcoxon signed rank test, Z=–1.68, p=0.093, n=8), indicating that the electrical stimulus delivered to the PBN at >10 Hz was as effective in recruiting the laryngeal motoneurons as activating fictive calling with 5HT or electrical stimulation.

To determine if the connections between the PBN neurons and the laryngeal motoneurons are monosynaptic, we repeated the experiments in high-divalent cation (Hi-Di) saline. We observed CAPs from both nerves in all animals in Hi-Di saline (n=7, *Figure 8D* middle column, F), but a higher stimulus amplitude was required to elicit CAPs in all cases (168+11.5% of the original stimulus amplitude). In addition, the amplitude of the CAPs was smaller in some cases (*Figure 8D and Hi–Di* column, 50 Hz), and sometimes pulses failed to evoke CAPs when stimulus frequency was >40 Hz (*Figure 8D* see Hi-Di column at 60 Hz, for example). The reduced responsiveness of the motoneurons to unilateral PBN stimulation is likely due to an increased spike threshold of neurons under the Hi-Di conditions (*Blitz and Nusbaum, 1999*). We, therefore, conclude that the synapse between the PBN and the laryngeal motoneurons are monosynaptic.

*Zornik and Kelley, 2008* previously showed that the synaptic transmission between the PBN projection neurons and the ipsilateral laryngeal motoneurons is mediated by glutamate and AMPA receptors. Here, we repeated this experiment to determine if the contralateral synapses between PBN projection neurons to the laryngeal motoneurons are also mediated by AMPA receptors by adding NBQX to the recording chamber. The results showed that the application of NBQX eliminated all PBN-evoked CAPs from both nerves (*Figure 8G*), indicating that glutamate and AMPA receptors mediate the synaptic transmission between the parabrachial neurons and the laryngeal motoneurons.

Next, we extended the experimental approaches to other fast clickers to characterize their synapses between the PBN and the laryngeal motoneurons. Unilateral PBN stimulation evoked CAPs from both nerves in 88% of the fast clickers (78% of male *X. amieti*, n=9, 100% of male *X. cliivi*, n=3, 100% of *X. petersii*, n=10, and 75% of testosterone-treated female *X. laevis*, n=4) (*Figure 9A*), with the remaining preparations showing CAPs from either the ipsilateral or contralateral nerve. The mean minimum frequency of stimulus train required to elicit CAPs was 13.8±2.56 Hz (mean ±s.e., n=26 including 8 male *X. amieti*, 3 male *X. cliivi*, 11 male *X. petersii*, and 4 T-female *X. laevis*), which did not differ from that of male *X. laevis* (Mann Whitney U test, Z=01.445, p=0.120), and did not differ significantly across species (ANOVA $F_{3, 22} = 0.541$, p=0.659). Similar to male, *X. laevis*, a stimulus train elicited a train of CAPs with progressive change in the amplitude and latency that could be fitted with exponential curves (mean ± s.e. cross-correlation coefficient of the fit = 0.784 ± 0.010 for CAP amplitude, 0.915+0.008 for CAP latency, *Figure 9B*, male *X. amieti* as an example). Mean $\tau$ for CAP amplitude potentiation and latency attenuation for all the fast clickers were 5.15±0.735 pulses, and 2.69±0.332 pulses (mean ± s.e.), respectively. The $\tau$ values for CAP amplitude and latency were frequency-independent in all species (*Table 1*), did not differ significantly from those of male *X. laevis* (Mann Whitney U test, Z=–1.69,–1.254, p=0.0918, 0.210 for amplitude and latency tau, respectively), and did not differ across species (ANOVA, $F_{3.17} = 1.75$, $F_{3.15} = 3.24$, p=0.195, 0.0521 for CAP amplitude and latency tau, respectively). Thus, in other fast clickers, the CAP amplitude reaches 95% of the maximum

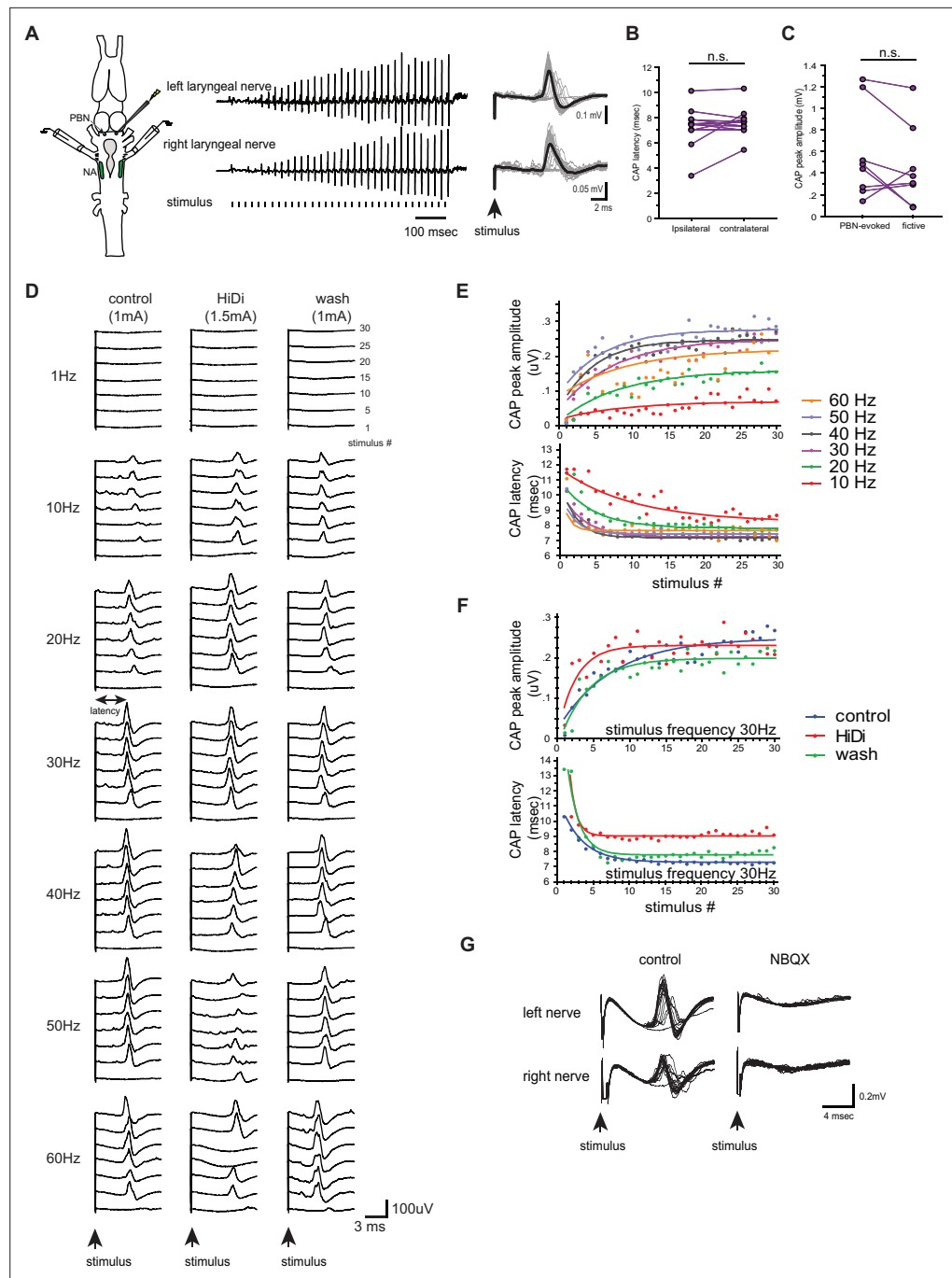

**Figure 8.** Properties of the synapses between the parabrachial nucleus (PBN) and laryngeal motoneurons in male *X. laevis*. (**A**). Left; A cartoon showing the dorsal view of the *Xenopus* brain showing the location of the stimulation and recordings. PBN; parabrachial nucleus, NA; nucleus ambiguus. Middle; recordings obtained from the left (top panel) and right (bottom panel) laryngeal nerves in response to the stimulus delivered to the right parabrachial nucleus (PBN). Tick marks in the bottom labeled as 'stimulus' indicate the timing of the stimulus pulse delivered to the PBN. Right, stimulus-evoked compound action potentials (CAPs) of the left (top panel) and right (bottom panel) laryngeal motor nerves shown as sweeps. Gray traces are raw data, and the thick black lines represent the mean response across 30 stimulus pulses. Arrow indicates the timing of stimulus pulse onset. (**B**). The latency between the stimulus onset and the time of the compound action potential (CAP) peak recorded from the ipsilateral and contralateral nerves. There is no significant difference in the CAP latency recorded from the two nerves, indicating the presence of bilateral projections from the PBN to the right and left laryngeal motoneurons. (**C**). The maximum amplitude of the CAPs evoked in response to the unilateral PBN stimulation and those

*Figure 8 continued on next page*

*Figure 8 continued*

observed during fictive advertisement calling. There was no significant difference in the maximum CAP amplitude recorded under these two conditions. (**D**). Frequency-dependence of the PBN-evoked CAPs in control (left), high-divalent cation saline (HiDi, middle), and after wash (right). Sweeps of laryngeal nerve response to the 1st, 5th, 10th, 15th, 20th, 25th, and 30th stimulus pulses delivered to the PBN (from bottom to top traces) distributed by 100uV for ease of visualization. Stimulus frequency ranged from 1 to 60 Hz. Note that consistent CAPs, except in response to the first pulse, were elicited when the stimulus frequency is above 10 Hz in control and after wash condition, but in the presence of HiDi saline, some CAPs are skipped. (**E**). The CAP peak amplitude (top) and the CAP latency (bottom) as a function of the stimulus pulse order when stimulus frequency ranged from 10 to 60 Hz. The progressive potentiation of the CAP amplitude and attenuation of CAP latency in response to each frequency were successfully fitted with exponential curves. (**F**). The CAP amplitude (top) and the CAP latency (bottom) as a function of stimulus pulse order in the presence and absence of HiDi saline. Stimulus trains at 30 Hz were delivered in control, in HiDi saline, and after wash. Even in HiDi saline, CAPs potentiated in amplitude and decreased in latency as in control and washout conditions, indicating that the synapses are monosynaptic. (**G**). Sweeps of left and right laryngeal nerve response to the unilateral PBN stimulation in control (left) and in the presence of NBQX (right).

within ~ 17 pulses, and the CAP latency reaches 95% of the minimum within ~ 8 pulses. As in male *X. laevis*, latency for CAPs evoked from the ipsi- and contralateral nerves were similar in all fast clickers (***Figure 9C***, Wilcoxon signed rank text, Z=–1.13, p=0.260, n=23 including 8 male *X. amieti*, 3 male *X. cliivi*, 9 male *X. petersii*, and 3 testosterone female *X. laevis*), indicating that bilateral projections from PBN to the right and left laryngeal motoneurons are common in fast clickers, enabling synchronous activation of the motoneurons from each PBN. Similar to male *X. laevis*, the maximum CAP amplitude achieved in response to unilateral PBN stimulation was similar to those recorded during fictive advertisement calls (***Figure 9D***, Wilcoxon signed rank test, Z=–1.16, p=0.245, n=16 including 5 male *X. amieti*, 3 male *X. cliivi*, 6 male *X. petersii*, and 2 testosterone-female *X. laevis*), indicating that unilateral stimulation of PB was effective in recruiting a comparable number of laryngeal motoneurons to those recruited during fictive vocal production.

We next repeated the experiment in Hi-Di saline and in NBQX. In Hi-Di saline, CAPs persisted in all animals (***Figure 9E and F***, n=6 male *X. amieti*, 2 *X. cliivi*, 5 *X. petersii*, and 3 T-treated female *X. laevis*), although increased stimulus amplitude was required to evoke CAPs in some brains (mean stimulus amplitude, 154+20% of control amplitude). In response to application of NBQX, PBN-evoke CAPs from both nerves were eliminated in all animals (n=3 male *X. amieti*, 2 male *X. cliivi*, 4 male *X. petersii*, 2 T-female *X. laevis*). These results indicate that the synapses between the PBN projection

**Table 1.** Parabrachial nucleus-evoked compound action potential amplitude and latency time constant $\tau$ in all fast clickers.

TF *X. laevis* stands for testosterone-treated female *X. laevis*. In response to a train of stimuli applied to a parabrachial nucleus (PBN) resulted in potentiation of the amplitude and the shortening of the latency of the nerve compound action potential (CAPs) that were successfully fitted by exponential curves $f(t) = A(1 - e^{t/\tau})^a C$ for each species (the second and third column). These $\tau$ values did not differ depending on the stimulus frequency based on the regression analysis in each species (the fourth and fifth column) with $\tau$ as dependent and stimulus frequency as independent variables.

| Sex/species | CAP amplitude τ by stimulus order (mean ±standard error, n) | CAP latency τ by stimulus order (mean ±standard error, n) | CAP amplitude τ by stimulus frequency regression | | CAP latency τ by stimulus frequency regression | |
|---|---|---|---|---|---|---|
| | | | F statistic | p-value | F statistic | p-value |
| Male *X. laevis* | 6.90±0.995, 11 | 3.64±515, 10 | $F_{1,24}$ = 0.113 | 0.739 | $F_{1,29}$ = 2.855 | 0.102 |
| Male *X. amieti* | 5.12±0.568, 6 | 3.48±0.708, 6 | $F_{1,25}$ = 3.152 | 0.088 | $F_{1,26}$ = 3.063 | 0.092 |
| Male *X. cliivi* | 5.01±1.61, 2 | 4.06±1.38, 2 | $F_{1,10}$ = 0.538 | 0.480 | $F_{1,10}$ = 0.377 | 0.553 |
| Male *X. petersii* | 3.84±0.700, 9 | 1.66±0.272, 7 | $F_{1,42}$ = 1.043 | 0.313 | $F_{1,29}$ = 0.185 | 0.185 |
| TF *X. laevis* | 8.23±3.20, 4 | 2.66±0.270, 4 | $F_{1,16}$ = 0.182 | 0.676 | $F_{1,18}$ = 0.2.37 | 0.141 |

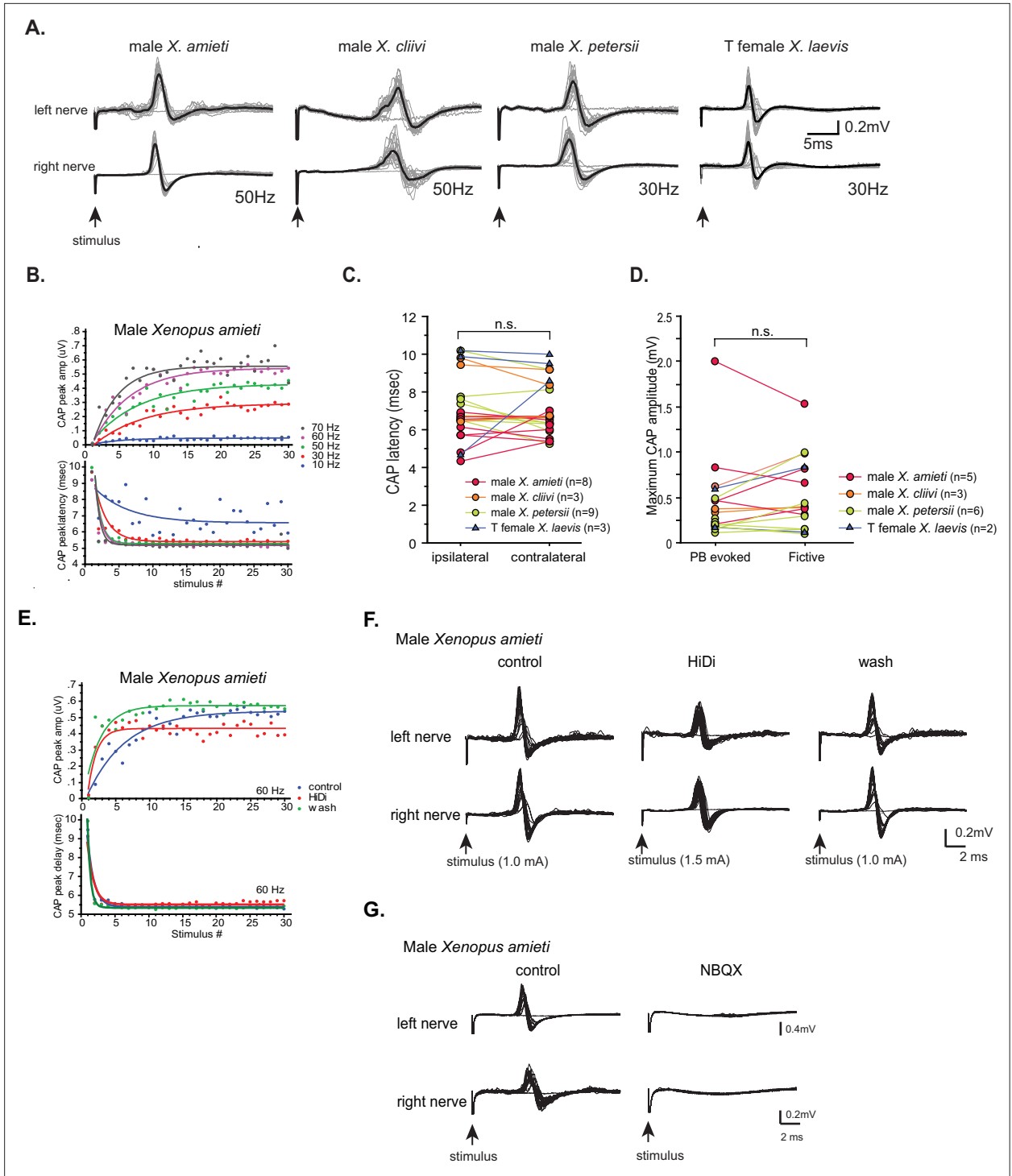

**Figure 9.** Properties of the synapses between the parabrachial nucleus (PBN) and laryngeal motoneurons in all fast clickers resemble those of male *X.laevis*. (**A**). PBN-evoked responses of the left (top panel) and right (bottom panel) laryngeal motor nerves shown as sweeps. Gray traces are raw data, and the thick black lines represent the mean response across 30 stimulus pulses. Arrows indicate the timing of stimulus onset, and the stimulus frequencies are shown below the traces. (**B**). PBN-evoked CAP amplitude (top graph) and the CAP latency (bottom graph) as a function of stimulus pulse order in response to stimulus trains (10–70 Hz) delivered to a PBN of a male *X. amieti* brain. Progressive increase in CAP amplitude and attenuation in CAP latency can be successfully fitted with exponential curves, as in male *X. laevis*. (**C**). PBN-evoked CAP latency recorded from the ipsilateral and contralateral nerve of all fast clickers. There was no significant difference in the latency between the two nerves. (**D**). Maximum CAP amplitude evoked in response to the unilateral PBN stimulation and those recorded during fictive advertisement calls. There was no significant difference in the maximum amplitude recorded under these two conditions. (**E**). Maximum CAP amplitude and the CAP latency as a function of the stimulus pulse number in

*Figure 9 continued on next page*

*Figure 9 continued*

control, HiDi saline, and washout in a male *X. amieti* brain. The progressive increase in CAP amplitude and the decrease in CAP latency were observed under all three conditions, indicating that they are monosynaptic. (**F**). Sweeps of left and right laryngeal nerves in response to the unilateral PBN stimulation in control (left), in the presence of HiDi saline (middle), and after wash. The arrows at the bottom show the timing of stimulus pulse delivery, with the stimulus frequency of 60 Hz in all three conditions. The amplitude used is shown in parenthesis. (**G**). Sweeps of left and right laryngeal nerves in response to the unilateral PBN stimulation at 60 Hz of male *X. amieti* in control (left), and in the presence of NBQX (right). The arrows at the bottom show the timing of stimulus pulse delivery.

neurons and bilateral laryngeal motoneurons of all fast clickers including testosterone-treated females are monosynaptic and mediated by glutamate and AMPA receptors. Thus, the synapses between the PBN and laryngeal motoneuron of all fast clickers share similar characteristics.

## All slow clickers have weak and unreliable synapses between the parabrachial nucleus and the laryngeal motoneurons

Lastly, we examined the property of parabrachial nucleus (PBN) to nucleus ambiguus (NA) synapses in slow clickers. Because the synapses are crucial in producing fast clicks (*Zornik and Yamaguchi, 2012*), we reasoned that the presence of synapses resembling those found in fast clickers in the brains of slow clickers provides evidence to support the idea that fast click central pattern generators (CPGs) are inherited, but remain latent in these species. When PBN of slow clickers were unilaterally stimulated, only about half of the brains responded with compound action pontentials (CAPs) from either one or both nerves (45% of female *X. laevis*, n=11, 57% of male *X. tropicalis*, n=7, *Figure 10A*). In the remaining preparations, there was either no response or tonic activity from the laryngeal nerves in response to PBN stimulation (*Figure 10B*). Even in the brains from which CAPs were elicited, some pulses within a stimulus train failed to evoke CAPs (*Figure 10A*), unlike in fast clickers where CAPs are elicited in response to almost every stimulus pulse (compare to male X. laevis *Figure 8D* left column). The mean minimum frequency required to elicit CAPs were >10.6 + 3.86 Hz in female *X. laevis*, and >30.0 + 8.17 Hz in male *X. tropicalis*. More importantly, in two female *X. laevis* brains and in one male *X. tropicalis* brain PBN needed to be stimulated at 20 Hz and 50 Hz, respectively, both of which are significantly higher than click rates of their vocalizations (female X. laevis 'rapping' contain clicks repeated at <20 Hz, and the advertisement call of male *X. tropicalis* contains clicks at ~30 Hz). Moreover, PBN-evoked CAP amplitudes of male *X. tropicalis* and female *X. laevis* were significantly lower and higher than those observed during fictive calling, respectively (male *X. tropicalis*: *Figure 10C*, left graph, Wilcoxon Signed rank test, Z=−2.02, p=0.043, female X. laevis: *Figure 10C*, Wilcoxon Signed rank test, Z=−2.02, p=0.043), indicating that PBN-evoked CAPs differ from those produced during fictive calling. The amplitude of PBN-evoked CAPs in slow clickers were significantly smaller than those of fast clickers (*Figure 10D*, ANOVA $F_{1, 50}$ = 8.766, p=0.0047), and the stimulus amplitude required to elicit CAPs from the laryngeal nerves was significantly higher in the slow clickers than in fast clickers (*Figure 10E*, ANOVA, $F_{1, 52}$ = 30.52, p<0.0001), indicating that synapses between the PBN and the laryngeal motoneurons of slow clickers are weaker than those of fast clickers. We did not determine whether the synapses were monosynaptic and glutamatergic because PBN-evoked CAPs in the slow clickers were variable and unreliable. Finally, we found that the properties of synapses between the PBN and the laryngeal motoneurons changed dramatically in response to testosterone in female *X. laevis*, suggesting that female *X. laevis* acquire fast trill-like CPGs in response to testosterone. To summarize, our results suggest that slow clickers lack the strong synapses between the PBN and the laryngeal motoneurons that are observed in the fast clickers.

## Discussion

Divergent evolution is the process by which closely related populations within a species accumulate different traits, which can eventually lead to speciation. In this study, we compared the central pattern generators (CPGs) that generate courtship vocalizations in five closely related species of *Xenopus*. We found that the fast trill-like and slow trill-like CPGs originally discovered in male *X. laevis* are not species-specific, but are inherited by fast and slow clickers, respectively (*Figure 11*). These CPGs are broadly tuned to produce clicks within a range of slow (6–35 Hz) and fast (50–150 Hz) rates. This observation is consistent with the idea that neuronal networks underlying behavior are well conserved

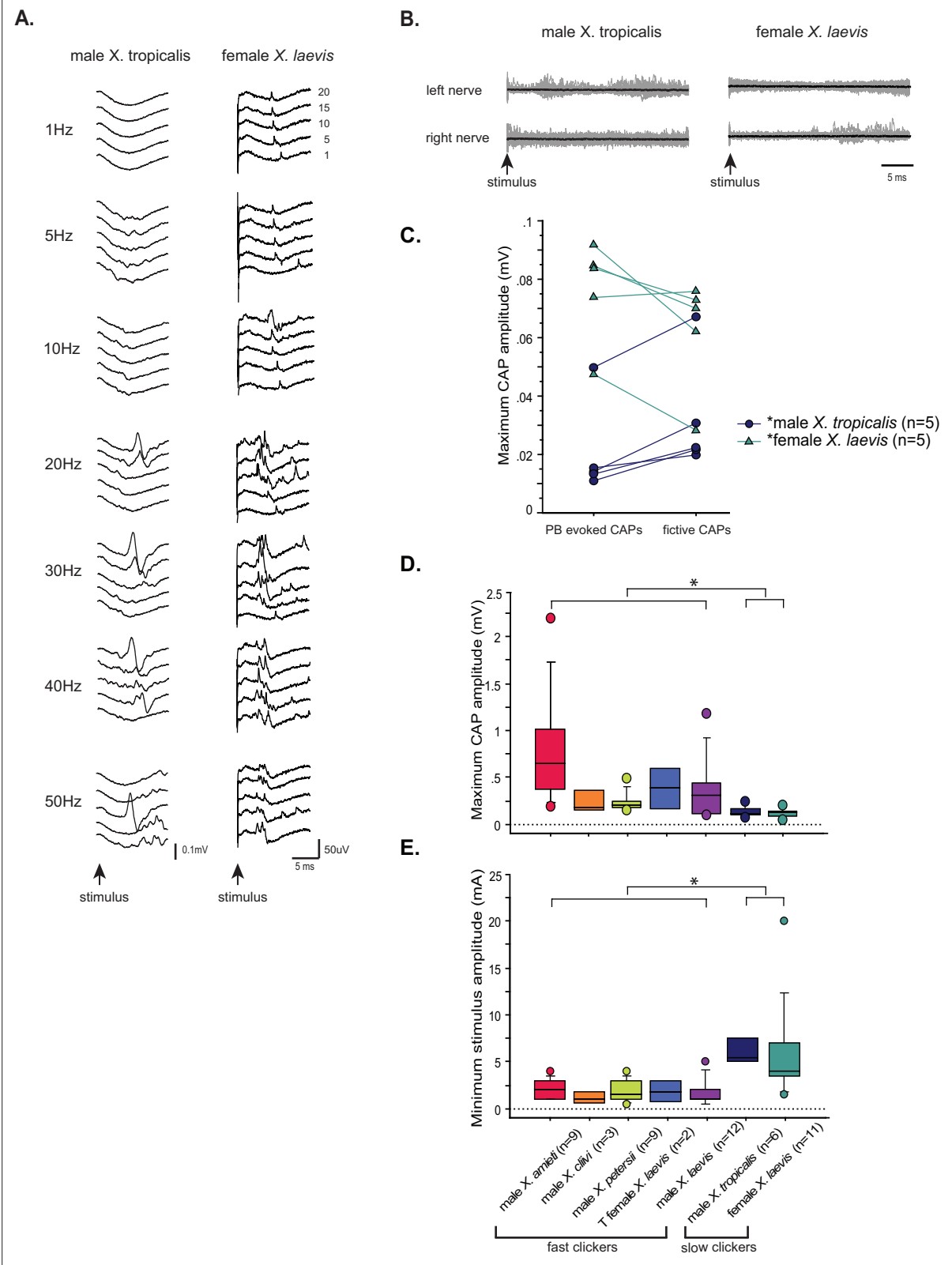

**Figure 10.** The synapses between the parabrachial nucleus (PBN) and laryngeal motoneurons of slow clickers are weaker and unreliable compared to those of fast clickers. (**A**). Example of slow clickers that showed compound action potentials in response to the unilateral stimulation of the PBN. Sweeps of laryngeal nerve responses to the 1st, 5th, 10th, 15th, and 20th stimulus pulses delivered to the PBN (from bottom to top traces) distributed by 50uV for ease of visualization. Stimulus frequency ranged from 1 through 50 Hz. (**B**). Example of slow clickers that showed no compound action potentials in

*Figure 10 continued on next page*

*Figure 10 continued*

response to the unilateral PBN stimulation. Sweeps of left (top) and right (bottom) laryngeal nerve recordings in response to stimulus pulses. Gray traces are raw data, and the thick black lines are mean responses. (**C**). Maximum CAP amplitude recorded in response to the unilateral PBN stimulation (PBN-evoked CAPs) and those observed during fictive calling (fictive CAPs) in male *X. tropicalis* (left), and in female *X. laevis*. In male *X. tropicalis* PBN-evoked CAPs were significantly smaller than those recorded during fictive advertisement calls, and in female *X. laevis*, PBN-evoked CAPs were significantly larger than those recorded during fictive release calls, as indicated by asterisks preceding the species name. (**D**). Box plot showing the maximum peak amplitude of PBN-evoked CAPs all species and sexes. The maximum peak amplitude of CAPs of the slow clickers was significantly smaller than those recorded from the fast clickers. (**E**). Minimum stimulus amplitude required to evoke CAPs in response to the unilateral PBN stimulation in all species and sexes. The slow clickers required a significantly higher amplitude of stimulus to evoke CAPs.

across species because they are generalists rather than specialists (*Katz and Harris-Warrick, 1999*). The finding implies that the central nervous system of male *Xenopus* can generate sufficient variation in click rates using the pre-existing CPGs. As a result, if female preference for the temporal structure of a courtship call evolves, variant males with matching click rates can be selected by sexual selection.

## Anatomically distinct organization of frequency-dependent CPGs, and their temporal coordination

Our results show that slow and fast clicks are generated by distinct central pattern generators (CPGs) across species; slow clicks repeated at rates <35 Hz are produced by slow trill-like CPGs contained in the caudal brainstem, whereas fast clicks (>50 Hz) are generated by fast trill-like CPG that span between the parabrachial nucleus (PBN) and nucleus ambiguus (NA). The results indicate that the basic architecture of the neural circuitry used to generate courtship advertisement calls appears to be conserved across species, even though the advertisement calls of each species differ in click repetition rates and the number of clicks within each call (in addition to the frequencies (i.e. pitch) of each click dictated by the anatomy of the larynx *Kwong-Brown et al., 2019*).

The use of different populations of neurons to generate rhythmic motor programs at different frequencies may be common in vertebrates. For instance, in the zebrafish swimming network and mice locomotory network, which varies greatly in oscillatory frequency, interneurons in the locomotory circuits are recruited in a rate-dependent manner (*McLean et al., 2007*; *McLean et al., 2008*; *Rancic et al., 2020*; *Song et al., 2020*). Specifically, in zebrafish, three different populations of interneurons that are recruited at slow, intermediate, and fast frequencies of undulations have been identified (*Song et al., 2020*), and similar results are reported in mice (*Rancic et al., 2020*). The recruitment of PBN neurons to generate fast, but not slow clicks in *Xenopus* may parallel the recruitment of fast-swim and slow-swim interneurons in zebrafish swim circuits. The use of different populations of interneurons to generate distinct frequency of motor programs driving the same muscle groups may be the most efficient strategy in vertebrates.

All the fast clickers included in this study use calls containing clicks repeated at a slow (<35 Hz) rate such as amplectant clicks, release calls, and long-slow clicks. In contrast, the slow clickers (male *X. tropicalis* and female *X. laevis*) never generate calls containing fast clicks (>50 Hz). This disparity in the frequency range of vocal repertoires may be due to differences in the effective organ of fast and slow clickers. Male *X. laevis* (fast clicker) laryngeal muscles have faster contractile properties compared to female *X. laevis* (slow clicker), corresponding with the differences in the click rates produced by each sex. The half-relaxation time of a single twitch of female laryngeal muscles is six times that of male laryngeal muscles (*Potter et al., 2005*), which means the female larynx is incapable of generating fast clicks even if the motoneurons fire at a high rate. To maintain fast twitch muscles, specialized mechanisms including increased expression of $Ca^{2+}$-ATPase by the sarcoplasmic reticulum are required to handle high levels of myoplasmic $Ca^{2+}$ (*Rome and Klimov, 2000*), which may come at a high cost. Therefore, if slow clicks are sufficient to fulfill the primary purpose of acoustic communication, such as attracting females, it may be cost-effective use slow clicks exclusively.

Biphasic callers, including male *X. laevis* and *X. petersii*, produce fast and slow trills that are mutually exclusive. Although coordination of multiple oscillators has been extensively studied in other species, neural mechanisms that coordinate the timing of fast and slow trills are yet to be determined. In lampreys, swimming oscillators in each segment of the spinal cord operate at same frequency generate undulatory motion (*Grillner et al., 1995*) and their coordination are explained by mathematical theory of phase-coupled oscillators are coupled with a fixed-phase difference to generate

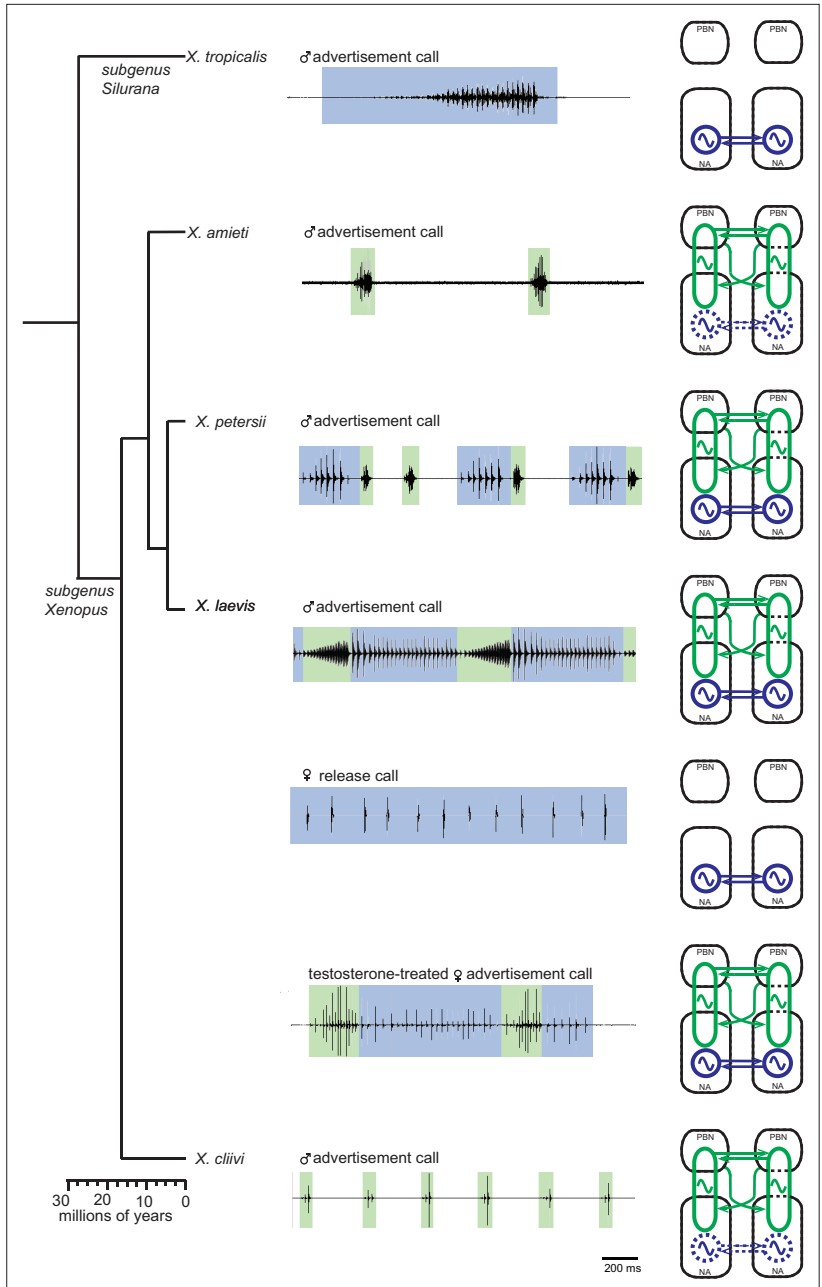

**Figure 11.** Summary of the two types of vocal central pattern generators with a conserved anatomical organization found in five species of genus *Xenopus*. Left column: chronogram based on mitochondrial DNA (Modified from *Evans et al., 2015*) of studied species. Middle column: fictive advertisement calls recorded ex vivo from the isolated brain. The green and blue background of traces indicate fast (>50 Hz) and slow (<35 Hz) clicks, respectively. Right column: schematics showing the types of central pattern generators (CPGs) thought to be contained in the brainstem of each species. A pair of parabrachial (PBN) and nucleus ambiguus (NA) are shown for each species, and the presence of slow CPGs (blue) and fast CPGs (green) are shown in each species, and in two sexes in the case of *X. laevis*. Dotted slow CPGs in X. amieti and X. cliivi indicate that their presence is yet to be tested in these species.

undulatory motion (***Marder et al., 1997***; ***Sigvardt and Miller, 1998***). Similarly, in crustaceans, two CPGs controlling chewing and food filtering that operate at two distinct frequencies modulate each other (***Weimann et al., 1991***; ***Thuma and Hooper, 2002***; ***Bucher et al., 2006***). In particular, synaptic connections between the two CPGs in Jonah crab (*Cancer borealis*) have been shown to coordinate rate and timing of both oscillators through fixed-latency coupling (***Bartos et al., 1999***). The

fast and slow CPGs of *Xenopus* operate alternately, which distinguishes them from the oscillators described above that operate simultaneously. Nonetheless, it is possible that reciprocal inhibition, a core building block of most CPGs (*Marder and Bucher, 2001*), may play a role in coordinating the timing of fast and slow CPGs. In the future, high-density recordings from the calling brain of biphasic species hold the potential to detect the presence of such neurons.

## Phylogeny of the vocal central pattern generators in the genus *Xenopus*

Determining which of the central pattern generators (CPGs) in the genus *Xenopus* is more ancestral is not straightforward because both fast and slow clickers are widely distributed across clades (*Tobias et al., 2011*; *Evans et al., 2015*). Studies on sound-producing fishes have shown that the neurons in the vocal CPGs in the caudal brainstem originate from rhombomere 8 (Bass and Chagnaud 2012). Although the exact location of the slow click CPGs in *Xenopus* is not known, it is possible that neurons derived from rhombomere 8 constitute the slow click CPGs in *Xenopus*, suggesting that the slow CPGs may be more ancestral than the fast CPGs. Although we did not test whether all fast clickers have slow CPGs in their brainstem in this study, they generate calls that consist of slow clicks, suggesting that slow CPGs may be inherited by all species of *Xenopus*.

In contrast, the parabrachial nucleus (PBN), which develops from rhombomere 0 and 1 (*Karthik et al., 2022*), is a part of the central vocal pathways (*Smotherman et al., 2006*; *Hage, 2020*) in mammals, and its contribution to vocal production is also suggested in birds (*Wild et al., 2009*). Given the involvement of PBN in coordinating vocal and respiratory systems (*Smotherman et al., 2006*; *Smotherman and Metzner, 2010*; *Barkan and Zornik, 2020*), its use is considered to be an adaptation in terrestrial species to power their vocalizations through respiration. Interestingly, the PBN is active during fictive advertisement calling in a terrestrial frog, *Rana pipiens* (*Schmidt, 1992*), suggesting that PBN may have been recruited into the central vocal pathways early in the evolution of tetrapod.

Although most species of frogs are terrestrial, *Xenopus* has returned to an aquatic life and produces vocalizations exclusively under water using a specialized larynx that does not require the respiratory system (*Yager, 1992*; *Kwong-Brown et al., 2019*). Air breathing occurs entirely at the water surface, making calling and breathing mutually exclusive in *Xenopus*. There is even some evidence of reciprocal inhibition between the two systems (Yamaguchi unpublished data). *Xenopus* likely inherited their fast CPGs, which include PBN, from ancestral terrestrial frog that used respiratory system to vocalize. They then modified these CPGs so that the vocal and respiratory activities occur in antiphase instead of in phase. Taken together, we speculate that *Xenopus* inherited the ancient pacemaker that existed in the caudal brainstem of fishes, which has evolved into slow click CPGs that may be shared by all species of Xenopus. Additionally, *Xenopus* likely inherited neural networks, including the PBN, from more recent terrestrial species that generate vocalizations using the respiratory system, which has evolved into fast click CPGs that are selectively shared by fast clicker species.

## Evolution of *Xenopus* fast click central pattern generators

In invertebrates, the divergent evolution of central pattern generators (CPGs) has been studied extensively (reviewed by *Sakurai and Katz, 2015*; *Katz, 2016*; *Roberts et al., 2022*). Results from these studies suggest that there arereveal three main strategies employed in this process: synaptic rewiring, changes in neuromodulation, and alteration in CPG neurons.

Synaptic rewiring involves changes in the connectivity of CPG neurons can result in behavioral divergence through evolutionary time. This process has been observed in various species including the pharyngeal pumping CPGs of nematodes (*Bumbarger et al., 2013*), the song CPGs of *Drosophila* (*Ding et al., 2019*) and crickets (*Jacob and Hedwig, 2019*; *Lin and Hedwig, 2021*), and has been implicated in the evolution of the unique locomotory CPGs of the Icelandic horse (*Andersson et al., 2012*). Molecular mechanisms underlying synaptic rewiring include modification of genes controlling axon guidance (*Andersson et al., 2012*) or synaptic development (*Charrier et al., 2012*; *Schmidt et al., 2021*), differential expression of surface molecules by synaptic target neurons (*Hong et al., 2012*), and the presence or absence of microRNA (*Johnston and Hobert, 2003*). It is important to note that synaptic rewiring can occur without changing the behavioral output, indicating that natural

selection can act upon the neural circuitry and motor outputs independently (*Sakurai et al., 2011*; *Sakurai et al., 2014*; *Katz, 2016*).

Neuromodulators can alter the output of the CPGs by altering the intrinsic properties of the constituent neurons and their synaptic connectivity (*Katz, 1998*; *Harris-Warrick, 2011*; *Miles and Sillar, 2011*; *Golowasch, 2019*; *Blitz, 2023*). Changes in the types, concentration, or the expression patterns of neuromodulators or receptors can all result in divergent behavioral output. For instance, the divergence of chewing and food filtering behavior generated by CPGs of two crustacean species is explained by changes in the type of neuromodulators (*Meyrand et al., 2000*).

Intrinsic properties of neurons that constitute CPGs can be modified by various mechanisms, including changes in the ionic conductance, receptor expression, and downstream signaling pathways (*Roberts et al., 2022*), and lead to divergent evolution. Divergence of key CPG neurons has been observed in the song/vocal CPGs of crickets (*Jacob and Hedwig, 2019*) and *Xenopus* (see below *Barkan et al., 2018*).

Although the exact location of slow click CPGs is not known, we have better understanding of the location and function of the fast click CPGs in *Xenopus*. However, we do not yet know whether synaptic wiring and changes in neuromodulations contribute to the divergence of fast clicks across species due to the insufficient resolution to detect subtle differences in synaptic wiring, and the use of a single type of neuromodulator (i.e. 5HT) to determine its impact on the CPG in this study. Nevertheless, we showed that, at a gross anatomical level, synapses between the parabrachial nucleus (PBN) and nucleus ambiguus (NA) are conserved across all fast clicker species. Additionally, we found that 5HT has species-specific effectiveness in activating fictive advertisement calls, indicating possible contribution of neuromodulators in the vocal divergence. In contrast, there is strong evidence that the intrinsic properties of a particular type of neuron in the fast CPG have diverged across species. Male *X. laevis*, *X. petersii*, and *X. victorianus* produce fast trills repeated at ~60 Hz with species-specific duration in their advertisement calls. Fast Trill Neurons identified in the parabrachial nucleus of these species fire at 60 Hz and exhibit long-lasting depolarization mediated by NMDA receptor activation, coinciding with the species-specific duration of fast trills (*Zornik and Yamaguchi, 2012*; *Barkan et al., 2017*; *Barkan et al., 2018*), indicating that the neurons code for the duration and click rates of fast trills in each species. Therefore, it is likely that fast click CPGs diverged across species by altering the intrinsic properties of key neurons in the parabrachial nucleus while conserving the basic synaptic wiring.

## Species-specific effects of testosterone on the configuration of fast trill-like central pattern generators

In this study, we found that when adult female *X. laevis* were treated with testosterone, their central vocal pathway transformed to resemble that of males. This included the activation of parabrachial nucleus (PBN) during fictive fast trills and the emergence of male-like synapses between PBN and laryngeal motoneurons. However, this effect of testosterone appears to be limited to the fast clicker species. Male *X. tropicalis*, a slow clicker species, has been shown to have comparable plasma levels of testosterone to male *X. laevis* (mean plasma levels of testosterone of male X. laevis: 13–22 ng/ml *Hecker et al., 2005*; *Hayes et al., 2010*, male X. tropicalis:~20 ng/ml *Olmstead et al., 2009*), yet the synapses between the PBN and laryngeal motoneurons in male *X. tropicalis* remained weak, and PBN showed little activity during fictive advertisement calls. These results suggest that testosterone acts differently on the central vocal pathways of fast and slow clickers, promoting the emergence of fast trill-like CPGs in *X. laevis* but not in *X. tropicalis*. Although further experiments with controlled testosterone levels are necessary to validate these results, we hypothesize that changes in the androgen receptors (e.g. expression patterns, ligand affinity) may have contributed to the divergence of fast and slow clickers. In line with this hypothesis, species-specific sensitivity to the gonadal hormones has been reported in a few species of *Xenopus* (*Leininger et al., 2015*).

## Fast trill-like central pattern generators are not latent in slow click species of *Xenopus*

Latent neural networks, which are considered evolutionary remnants, have been observed in various animals. For example, female *Drosophila melanogaster*, which do not produce male-typical courtship songs, still possess song neural networks that can be artificially activated (*Clyne and Miesenböck,*

*2008*). Similarly, flight neural networks are present in flightless grasshoppers (*Arbas, 1983*). Given that fast trill-like CPGs in the genus *Xenopus* are likely ancestral, wIn this study, we asked if the fast trill-like CPGs are inherited by all species within the genus *Xenopus*, despite being latent in slow clickers. Although we did not observe any fictive fast clicks from the brains of male *X. tropicalis* and female *X. laevis,* we could not rule out the possibility that the motor programs were not elicited because we failed to provide appropriate stimuli. To address this question, we examined the synaptic properties between the parabrachial nucleus and the laryngeal motoneurons, which are crucial for the fast trill-like central pattern generators (CPGs). Our results revealed that fast and potentiating synapses found in all fast clickers are absent in slow clickers. Instead, the synapses of slow clickers are weak and unreliable. Thus, we conclude that fast trill-like CPGs are present only in species that incorporate fast clicks in their vocal repertoire, possibly due to the high cost of maintaining such synapses (*Figure 11*). If slow clickers produce fast-trill like motor programs, which is unlikely due to the limitation of the laryngeal contractile properties discussed above, the underlying neural circuitries likely differ from the fast trill-like CPGs. It is important to note that the addition of fast trill-like CPGs to the central vocal pathways can occur rapidly within an individual, as seen in the transformation of the central vocal pathways in adult female *X. laevis*.

In summary, we compared the central vocal pathways of five *Xenopus* species to better understand neural mechanisms underlying divergence of courtship behavior. We found that fast and slow trill-like CPGs have a conserved basic architecture across species. Fast trill-like CPGs are not shared by all *Xenopus* species, but present only in fast clickers. Exogenous testosterone appears to promote the acquisition of fast trill-like CPGs in fast clickers, but not in slow clickers. We suggest that changes in the role of testosterone in organizing the central vocal pathways resulted in the divergence between fast and slow clickers, and the alteration of the intrinsic properties of Fast Trill Neurons in the parabrachial nucleus led to species-specific rate and duration of clicks among fast clickers. Our findings indicate that courtship behavior in vertebrates can diverge across species even though the underlying neural circuits inherited through an evolutionary lineage are conserved. This behavioral divergence is likely achieved through species-specific modifications in the intrinsic properties of neurons, and potential changes in responsiveness to gonadal hormones.

# Materials and methods
## Animals
Fifty-nine male *X. laevis*, 39 female *X. laevis*, 24 male *X. tropicalis* obtained from Nasco (Fort Atkinson, WI, average +std weight = 42.59 ± 3.04 g, 54.83±4.79 g, 10.13±0.32 g, length = 6.99 ± 0.18 cm, 7.63±0.20 cm, 4.48±0.05 cm), 13 male *X. petersii* obtained from the National *Xenopus* Resource (Woods Hole, MA, weight = 16.49 ± 0.70, length = 4.22 ± 0.09 cm), and 12 male *X. amieti*, 3 male *X. cliivi* generously provided by Dr. Darcy Kelley (Columbia University, weight = 7.24 ± 1.23 g, 16.4±0.144 g, length = 3.87 ± 0.2 cm, 5.33±0.2 cm) were used for this study. This study was performed in strict accordance with the recommendation in the Guide for the Care and Use of Laboratory Animals of the National Institute of Health. All the animals were handled according to approved institutional animal care and use committee (IACUC) protocols (#00001989) of the University of Utah. The protocol was approved by the Institutional Animal Care and Use Committee at the University of Utah and complied with National Institute of Health guidelines. All surgery was performed under tricaine methanesulfonate (MS-222) anesthesia, and every effort was made to minimize suffering.

## Gonadectomy and testosterone implants
Eleven adult female *X. laevis* (average + std weight=64.39 + 6.43 g, length = 8.08 + 0.26 cm) were anesthetized with MS-222 (ethyl 3-amino benzoate methanesulfonic acid, Sigma Aldrich) and were ovariectomized by removing ovaries and fat bodies using a cauterizer through a small incision (~1 cm) in the abdomen. Immediately after ovary removal, testosterone-filled (4-androsten-17β-ol-3-one, Sigma Aldrich T-1500) Silastic tubes (2.16 OD x 1.02 ID; 0.5 mg/g body weight) were implanted into the ventral lymph sacs of females. This treatment is known to elevate plasma levels of testosterone from female-typical levels (1.1–2.3 ng/ml, *Kang et al., 1995*) to levels about 70% higher than those of reproductively receptive males (44 ng/ml) for >1 year (*Watson and Kelley, 1992*; *Kang et al., 1995*). After 13 weeks, vocal recordings were obtained from the testosterone-treated female *X. laevis*

(*Potter et al., 2005*). Once they were verified to produce male-like advertisement calls, the animals were used for electrophysiological experiments.

## Sound recordings

Eight male *X. tropicalis*, 5 female *X. laevis*, 6 male *X. amieti*, 5 male *X. laevis*, 5 testosterone-treated female *X. laevis*, 3 male *X. cliivi*, and 6 male *X. cliivi* were used for vocal recordings. Sound recordings were obtained from all animals with hydrophones (Aquarian Audio Hydrophone H1a, Anacortes, WA) suspended 2 cm below the surface level in the center of a plastic 12 liter tank filled with 10 liters of water. Vocal recordings were obtained using a voice-activated recording system (Sound Analyses Pro, http://soundanalysispro.com/). Advertisement calls of the males of all species and testosterone-treated females were recorded when housed individually. To facilitate vocal production in female *X. laevis* that did not produce release calls when housed solo, they were co-housed with male *X. laevis* (the vocalizations of the two sexes can be easily distinguished based on the click rate and the sound frequency of the clicks). All sound recordings were obtained at 19–22 $^0$C in the dark overnight.

## Sound analysis

Ten sound files were sampled randomly for analysis from each animal. To characterize the temporal morphology of the calls, the inter-click interval, and the number of clicks per call (or fast and slow trill vocal phases in the case of male *X. laevis*, *X. petersii*, and testosterone-treated female *X. laevis*) were measured using Raven software (Cornell University Bioacoustics Laboratory, Ithaca, NY). To characterize the central tendency of click rates for each individual, we carried out the following analysis. First, normalized frequency histograms of instantaneous click rates (1/inter-click interval in seconds) were plotted with a bin width of 1 Hz for each individual. Each histogram was fitted with either a unimodal (for animals that produce monophasic calls; male *X. amieti*, male *X. cliivi*, male *X. tropicalis*, and female *X. laevis*) or a bimodal (for animals that produce biphasic calls; male *X. laevis*, male *X. petersii*, and testosterone-treated female *X. laevis*) Gaussian distribution using the Levenberg-Marquardt method to search for a model and the sum of squared errors method to identify the model that best fit the data. The mean value for μ (for monophasic calls), or μ1 and μ2 (for biphasic calls) were used to represent the central tendency of click rates of each individual.

## Isolated brain preparation

The methods for isolating a brain were described elsewhere (*Rhodes et al., 2007*; *Zornik and Yamaguchi, 2012*). Briefly, animals were anesthetized with subcutaneous injection (0.3 mL 1.3%) of tricaine methanesulfonate (MS-222; Sigma), decapitated on ice, and brains were removed from the skulls in a dish containing cold saline (in mM: 96 NaCl, 20 NaHCO$_3$, 2 CaCl$_2$, 2KCl, 0.5 MgCl$_2$, 10 HEPES, and 11 glucose, pH 7.8 oxygenated with 99% O$_2$/1% CO$_2$). Brains were then brought back to room temperature (22 °C) over the next hour and then transferred to a recording chamber which was superfused with oxygenated saline at 100 ml/hour at room temperature.

Fictive calls were recorded from 8 male *X. tropicalis*, 7 female *X. laevis*, 4 *X. amieti*, 5 male *X. laevis*, 3 testosterone-treated female *X. laevis*, 2 male *X. cliivi* and 5 male *X. petersii*. After fictive calls were recorded from intact brains, unilateral transections between the PBN and the NA on either the left or the right side were made using a scalpel in some brains. To prevent the neurons from excessive firing during transection, brains were first chilled by superfusing with ice-cold saline to bring the temperature of the tissues to ~5 °C, and a cut was made posterior to cranial nerve VIII from the midline to the lateral edge of the brains on either the left or the right side of the brainstem. After the transection, the brains were gradually warmed up by superfusing with room-temperature saline at 100 ml/hr. To confirm the completeness of the transection, Texas red dextran and fluorescein dextran (both 3000 mw, Thermo Fisher, Waltham, MA) were deposited into the nucleus ambiguus (NA) on each side using minutien pins after the completion of the electrophysiological recordings, and the brain was incubated at 4 °C for 48 hr, fixed in 4% paraformaldehyde, sectioned into 40 um thickness, and examined under the fluorescent microscope to verify that the no labeled soma or axons are seen in the PBN of the transected side.

## Electrophysiology

Extracellular recordings of the left and right laryngeal nerves were obtained using suction electrodes placed over the cranial nerve IX-X. Signals were amplified (1000 X) and bandpass filtered (10–10 kHz) with a differential A-C amplifier (model 1700, A-M systems). Local field potential (LFP) recordings from the parabrachial nucleus (PBN) were obtained using a 1 MΩ tungsten electrode (FHC, Bowdoin, ME) with its signal amplified (1000 X) and bandpass filtered (0.1–10 kHz) using an amplifier (model 1800, A-M systems). Both nerve and PBN LFP recordings were digitized at 10 kHz (Digidata 1440 A; Molecular Devices, San Jose, CA) and acquired with Clampex software (Molecular Devices).

Fictive calls were elicited using the following methods. For male, female, testosterone-treated female *X. laevis*, and male *X. petersii*, the bath application of 60 µM serotonin (5-HT, Sigma-Aldrich,) was effective in eliciting fictive calls. For male *X. tropicalis*, the bath application of 30 µM 5-HT together with 50 µM N-methyl-DL-aspartic acid (NMA, Sigma-Aldrich) was applied to the isolated brains. For male *X. amieti* and male *X. cliivi*, electrical pulses were delivered to either the left or right rostral-lateral cerebellum (RLCB, *Figure 1D*) via a concentric electrode (CBPH75, FHC, Bowdoin, ME) connected to an SIU (Iso-Flex, A.M.P.I, Jerusalem, Israel) driven by a stimulator (Master 8, A.M.P.I, Jerusalem, Israel) to elicit fictive advertisement calls. For the electrical stimulation, the stimulus pulse duration was 40 µs in all cases, but the pulse frequency, amplitude, polarity, and the total number of pulses per train necessary to elicit fictive calls were empirically determined for each preparation. Electrical stimulation of RLCB was also used to elicit fictive calls in some of the brains of other species in this study.

To functionally characterize the synapses between the NA projecting neurons in PBN and the laryngeal motoneurons, a 1 MΩ tungsten electrode (FHC, Bowdoin, ME) was inserted into either the left or right PBN and stimulus trains (40us pulse duration, frequency ranged from 1 to 100 Hz) were delivered via an SIU (Iso-Flex, A.M.P.I, Jerusalem, Israel) driven by a stimulator (Master 8, A.M.P.I, Jerusalem, Israel) while recordings are obtained from the left and right laryngeal nerves as described above. The minimum stimulus amplitude necessary to elicit compound action potentials was empirically determined first for each preparation. In some animals, the experiments were repeated in high-divalent cation (Hi-Di) saline to determine if the synaptic connections are monosynaptic. Hi-Di saline is a commonly used method to remove polysynaptic components to isolate monosynaptic components of the synapses (*Nicholls and Purves, 1970*). Hi-Di saline included four times the concentration of $MgCl_2$ (2 mM) compared to regular saline (0.5 mM), with the concentration of NaCl adjusted to maintain the constant osmolarity. To determine the role of AMPA receptors in mediating the synaptic transmission between PBN and laryngeal motoneurons, in some animals the PBN stimulation experiment was repeated in a bath with 2 µM NBQX added to the saline in some animals.

## Electrophysiological data analyses

The local field potential (LFP) recordings obtained from the PBN contain phasic activity that correlated with >50 Hz compound action potentials (CAPs) recorded from the laryngeal nerve during CAPs repeated at rates >50 Hz. We used the power spectral density (PSD) of the LFP traces (Clampfit software, Molecular Devices). To calculate the mean PSD for each species, each PSD was normalized to its maximum value and averaged across individuals from recordings of the same length (to hold the spectral resolution constant) within each species.

The *Xenopus* larynx generates a click sound when both laryngeal muscles are activated simultaneously and pull apart a pair of arytenoid discs (*Yager, 1992*; *Kwong-Brown et al., 2019*). Accordingly, the central vocal pathways of intact *Xenopus* brains activate left and right laryngeal motoneurons synchronously. To quantify the synchronicity of the CAPs recorded from the left and right laryngeal nerves, cross-correlation coefficients between the two nerve recordings were calculated while sliding one nerve recording against the other across time (±10ms, except in female *X. laevis* where ± 50ms was used to accommodate their longer CAP duration). Specifically, 10 consecutive CAPs recorded during a fictive call from the left and right nerve of a brain were used to calculate cross-correlation coefficients, and the time at which the maximum cross-correlation coefficients was achieved was identified as 'the maximum lag time' for each brain. For biphasic callers, the maximum lag times were obtained for both fast and slow trills. A maximum lag time of zero indicates the synchronous activity of the two nerves. A maximum lag time value > 0 indicates a delay of the transected side relative to the intact side.

## Statistical analyses

To determine if the temporal morphology of the vocalizations recorded in vivo differs significantly from the fictive vocalizations recorded in vitro, we compared the click/conmpound action potential (CAP) rates and the total number of clicks/CAPs per call/vocal phase (fast or slow trills) using a Wilcoxon signed rank test for each species.

To assess the synchronicity of CAPs produced by the left and right nerves in intact brain preparations, we used a one-sample sign test to determine if the maximum lag time differs from 0. To assess if CAPs desynchronize after brain transection, we used a Wilcoxon signed rank test to compare the maximum lag time before and after transection.

Unilateral stimulation of the parabrachial nucleus (PBN) elicited CAPs from both laryngeal nerves. When the connection between the PBN and its ipsilateral nucleus ambiguus (NA) was transected, stimulation delivered to the PBN on the intact side still elicited CAPs from both nerves. To determine if the latency between the stimulus onset and the peak CAP time is significantly increased after the unilateral transection, we compared the CAP latency (defined as the interval between the stimulus onset and the time of CAP peak) before and after the transection on the intact side and to the transected side using the Mann-Whitney U test.

Unilateral stimulation of the PBN elicited CAPs from both laryngeal nerves in most animals. The significance of the difference in latency between stimulus onset and peak CAP times recorded from ipsilateral and contralateral nerves was evaluated using the Wilcoxon signed-rank test was used. In addition, to model changes in CAP amplitude and latency in response to repetitive unilateral stimulation of the PBN, both variables were plotted against the stimulus pulse number, and exponential curves $f(t) = A\,(1-e^{-t/\tau})^a + C$ were fitted for every stimulus frequency above 10 Hz for each individual with Chebychev search method and sums of squared errors minimization method using Clampfit software (Molecular Devices, San Jose, CA). An ANOVA was used to determine if $\tau$ differs depending on the stimulus frequency. In all animals that showed CAPs in response to PBN stimulation, the maximum amplitude of the PBN-evoked CAPs was compared to those recorded during fictive calling using a Wilcoxon signed rank test. The minimum stimulus amplitude required for PBN stimulation to elicit CAPs and the maximum amplitude of PBN-evoked CAPs were compared across species using the Mann-Whitney U test. All statistical tests were carried out using StatView software (SAS Institute, Cary, NC).

## Acknowledgements

We thank Akemi Nguyen for the histological analyses, Michelle Tin, and Berlyn Prue for the help with data analyses, and Darcy Kelley, Erik Zornik, Matt Wachowiak, Paul Katz, and Francois Lambert for the comments on the earlier version of the manuscript. This work was supported by NSF IOS-1934386 (AY).

## Additional information

### Funding

| Funder | Grant reference number | Author |
| --- | --- | --- |
| National Science Foundation | IOS 1934386 | Ayako Yamaguchi |

The funders had no role in study design, data collection and interpretation, or the decision to submit the work for publication.

### Author contributions

Ayako Yamaguchi, Conceptualization, Data curation, Formal analysis, Supervision, Funding acquisition, Validation, Visualization, Methodology, Writing - original draft, Project administration, Writing - review and editing; Manon Peltier, Data curation, Formal analysis

### Author ORCIDs

Ayako Yamaguchi http://orcid.org/0000-0002-5653-1041

## Ethics

This study was performed in strict accordance with the recommendation in the Guide for the Care and Use of Laboratory Animals of the National Institute of Health. All the animals were handled according to approved institutional animal care and use committee (IACUC) protocols (#00001989) of the University of Utah. The protocol was approved by the Institutional Animal Care and Use Committee at the University of Utah and complied with National Institute of Health guidelines. All surgery was performed under tricaine methanesulfonate (MS-222) anesthesia, and every effort was made to minimize suffering.

## Decision letter and Author response

Decision letter https://doi.org/10.7554/eLife.86299.sa1
Author response https://doi.org/10.7554/eLife.86299.sa2

---

## Additional files

### Supplementary files
• MDAR checklist

### Data availability

The data used to obtain the results of this article have been deposited on Dryad and can be viewed via https://doi.org/10.5061/dryad.2280gb5x3.

The following dataset was generated:

| Author(s) | Year | Dataset title | Dataset URL | Database and Identifier |
|---|---|---|---|---|
| Yamaguchi A, Peltier M | 2023 | Electrophysiological data from five species of *Xenopus* | https://dx.doi.org/10.5061/dryad.2280gb5x3 | Dryad Digital Repository, 10.5061/dryad.2280gb5x3 |

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
