## [Editor Report]

This important paper compares the neural basis for different calling songs in five species of clawed *Xenopus* frogs using neural activity recordings combined with lesions of pathways and stimulation of specific parts of the circuit. The evidence supporting the claims is solid and reveals conservation and variation in the circuits generating fast and slow clicks in courtship calls. The work will be of broad interest to neurophysiologists beyond the vocalization topic.

---

## [Decision Letter]

**Decision letter after peer review:**

Thank you for submitting your article "Two conserved vocal central pattern generators broadly tuned for fast and slow rates generate species-specific vocalizations in *Xenopus* clawed frogs" for consideration by *eLife*. Your article has been reviewed by 2 peer reviewers, and the evaluation has been overseen by a Reviewing Editor and Claude Desplan as the Senior Editor. The following individuals involved in the review of your submission have agreed to reveal their identity: Paul Katz (Reviewer #1); Francois M Lambert (Reviewer #2).

*Reviewer #1 (Recommendations for the authors):*

The protocols to elicit a fictive call differed. Sometimes 5HT was sufficient, sometimes, electrical stimulation, and in one species 5HT plus NMA was needed. This may raise concerns that the only reason that the fast trill was not observed in all species was that the correct stimulus was not found.

Figure 1, the two-color drops and lightning bolts are not as immediately informative as the text would be. They require the reader to look up what the symbols mean. Worse, the pink and the purple and not easily distinguished. Just add subheadings to those parts of the figure "5-HT", "5HT and NMA" or "Electrical stimulation" The example shown was produced by just one means. Although it is true that either 5HT or electrical stimulation could produce this activity, only one did. This is a new result and therefore needs to be documented. A supplemental figure could show a comparison of serotonin and electrical stimulation. This is also an issue for Figure 4E

There were many points in which the writing was awkward and could be improved.

For example, the rhetorical question in the introduction, "How do the neural circuits underlying species-specific courtship behaviour evolve?" could be replaced with a declarative sentence of intent.

The aspect of the introduction was poorly worded and do not set the paper up well.

For example the phrase, "we examined if…" appears twice. One can determine if something is true, but to examine something is not to make a determination of truth.

Similarly "we analyzed the function of the neural circuitry using ex vivo preparations using four species with courtship calls containing clicks at variable rates." The function the circuit is to produce the clicks, so the authors did not analyze the function, maybe they analyzed the mechanism underlying this function.

p. 6, line 19. Referencing "(Figure 2A fourth from the top panel)" suggests the need for sub-panel numbers (i, ii, iii, iv).

p. 6, line 21. What is "large activity"? Activity is not a quantity that is small or large. The authors should find a metric to quantify this or categorize it.

Figure 2. The x- axis needs to be labelled in B, not just C.

p. 8, line 9. "smaller role" suggests a comparison with an advertisement call, but it should be made explicit in the sentence.

P. 8, lines 16-18. It is awkward to have questions inserted into a paragraph of results.

P. 11, line 6, change to "of 2 testosterone-treated female *X. laevis*"

Figure 5. Some of the cross-midline connections are represented as double arrows and others as two distinct arrows. It is not clear why they are different.

p. 14, line 8, and p. 27, line 24, "high-divalent (Hi-Di) saline" should be changed to high-divalent cation (Hi-Di) saline.

I think that the slow and fast CPGs are more analogous to the way that the pyloric and gastric mill CPGs interact in the STG than different speeds of spinal locomotion, which can only do one thing at a time.

Here are some citations for it:

- Bartos M, Manor Y, Nadim F, Marder E, Nusbaum MP. Coordination of fast and slow rhythmic neuronal circuits. Journal of Neuroscience. 1999;19(15):6650-60.

- Bucher D, Taylor AL, Marder E. Central pattern generating neurons simultaneously express fast and slow rhythmic activities in the stomatogastric ganglion. J Neurophysiol. 2006;95(6):3617-32.

The discussion could benefit from a more complete review of what is known about comparative circuits, particularly in invertebrates, where the CPGs are well established. There are several reviews on this that could be cited. Here are two:

- Roberts RJV, Pop S, Prieto-Godino LL. Evolution of central neural circuits: state of the art and perspectives. Nature reviews Neuroscience. 2022;23(12):725-43.

- Katz PS. Evolution of central pattern generators and rhythmic behaviours. PhilosTransRSocLond B BiolSci. 2016;371(1685):20150057.

*Reviewer #2 (Recommendations for the authors):*

This study by Yamaguchi and Peltier provides a detailed investigation of the brainstem CPG functional organization that rules vocal behaviors in *Xenopus* species, from an evolutionary perspective. The main conclusion of the paper reveals that vocal CPGs, located in the brainstem, generating fast and slow clicks in Xenopus male courtship calls are conserved across various Xenopus species. But the development of the fast CPG depends on testosterone only in species producing fast-click courtship calls.

I did really appreciate reviewing this paper even if I'm not an expert in vocalization and involved neuronal networks. It's a real pleasure to read a manuscript submitted to a valuable journal like Elife that proposed clear results based on classical but robust and established methods to answer clear scientific questions with the appropriate model. At least, but unfortunately too few, it's the proof, that there is no need for so-called "cutting edge" novel technics and fashion transgenic models to produce valuable sciences. Actually, using in vitro neurophysiology to compare the functional organization of the same neuronal circuit across species is a real "tour de force" project for a laboratory. Moreover, such type of study that explores adaptive mechanisms in neuronal networks through their evolution to ensure a similar behavior constitutes an exciting question that is too few promoted in modern neurosciences. I truly think this paper represents a valuable study, that will inspire researchers beyond the vocalization field. I recommend this paper for publication in eLife.

The manuscript is well-written and easy to follow in general. I don't have major comments

---

## [Author Response]

Reviewer #1 (Recommendations for the authors):The protocols to elicit a fictive call differed. Sometimes 5HT was sufficient, sometimes, electrical stimulation, and in one species 5HT plus NMA was needed. This may raise concerns that the only reason that the fast trill was not observed in all species was that the correct stimulus was not found.

We believe that the crux of the matter is whether male *X. tropicalis* and female *X. laevis* have fast CPGs that we were unable to activate ex vivo because we did not provide appropriate stimulus. We cannot rule out the possibility of a fast trill-like motor program being elicited from a slow clicker brain under a correct context. However, we indirectly addressed this question by examining the synapses between the parabrachial projection neurons and the laryngeal motoneurons, which are crucial for the function of fast CPGs. Our results showed that the synapses of male *X. tropicalis* and female *X. laevis* differ systematically from those of fast clickers. The findings suggest that even if slow clickers possess the neural circuitry capable of producing fast clicks, synapses between the parabrachial nucleus and the laryngeal motoneurons are unlikely to be part of the circuitry. Therefore, we concluded that male *X. tropicalis* female *X. laevis* do not have fast CPGs comparable to those seen in fast clickers. This point is now included in the Discussion section titled “Fast trill-like central pattern generators are not latent in slow click species of *Xenopus*” (p27, ln 8 – 16).

Figure 1, the two-color drops and lightning bolts are not as immediately informative as the text would be. They require the reader to look up what the symbols mean. Worse, the pink and the purple and not easily distinguished. Just add subheadings to those parts of the figure "5-HT", "5HT and NMA" or "Electrical stimulation" The example shown was produced by just one means. Although it is true that either 5HT or electrical stimulation could produce this activity, only one did. This is a new result and therefore needs to be documented. A supplemental figure could show a comparison of serotonin and electrical stimulation. This is also an issue for Figure 4E

The changes are made in Figure 1 and 4 following the suggestion.

Following the advice, we expanded the Results section titled “Isolated male brains of all species generate fictive advertisement calls resembling in vivo calls when exposed to appropriate stimuli” to provide a more detailed description of the stimuli that we used to elicit fictive calls. Additionally, we have added Figure 1—figure supplement1, which shows example responses elicited by multiple stimulus types. The figure highlights that in male *X. tropicalis* that generate fictive advertisement calls in response to a combination of 5HT and NMA, while 5HT alone or electrical stimulation to the rostral lateral cerebellum (RLCB) failed to elicit a fresponse. In male *X. amieti,* fictive calls are generated in response to electrical stimuli delivered to RLCB, but not in response to application of 5HT. Lastly, in male *X. laevis,* fictive calls are generated in response to both 5HT and electrical stimuli delivered to the RLCB.

There were many points in which the writing was awkward and could be improved.For example, the rhetorical question in the introduction, "How do the neural circuits underlying species-specific courtship behaviour evolve?" could be replaced with a declarative sentence of intent.The aspect of the introduction was poorly worded and do not set the paper up well.For example the phrase, "we examined if…" appears twice. One can determine if something is true, but to examine something is not to make a determination of truth.Similarly "we analyzed the function of the neural circuitry using ex vivo preparations using four species with courtship calls containing clicks at variable rates." The function the circuit is to produce the clicks, so the authors did not analyze the function, maybe they analyzed the mechanism underlying this function.

In addition to replacing the sentences and words listed by the reviewers, we rewrote the manuscript to eliminate awkward wording and improve readability.

p. 6, line 19. Referencing "(Figure 2A fourth from the top panel)" suggests the need for sub-panel numbers (i, ii, iii, iv).

Sub-panel numbers have been added to Figure 2A.

p. 6, line 21. What is "large activity"? Activity is not a quantity that is small or large. The authors should find a metric to quantify this or categorize it.

In the revised manuscript, we have replaced the ambiguous terms “large” and “small” with a more specific description of the PBN activity during fictive fast and slow trills. Specifically, we have stated that during fictive fast trills, the PBN is active, and during fictive slow trills, the PBN is either inactive or exhibits minimal activity that does not exceed twice the amplitude of the noise (p8, ln 3 -5)

Figure 2. The x- axis needs to be labelled in B, not just C.

The correction has been made.

p. 8, line 9. "smaller role" suggests a comparison with an advertisement call, but it should be made explicit in the sentence.

We have revised the sentence to provide more clarity regarding the role of the PBN in producing fictive slow and fast clicks. The updated sentence reads as follows: ‘These results suggest that, regardless of vocal repertoire of the species, the PBN does not play a significant role, if any, in producing fictive slow clicks (< 35Hz) compared to its role in producing fast clicks (>50Hz).’ (p9, ln 19 – 21).

P. 8, lines 16-18. It is awkward to have questions inserted into a paragraph of results.

The sentence has been revised to clarify the objectives of the study and avoid the insertion of questions. The revised sentence reads as follows: ‘Our objective was to determine whether female *X. laevis* use slow trill-like CPGs, similar to those observed in male X. laevis, to produce release calls. Additionally, we aimed to determine if females acquire fast trill-like CPGs or continue to use the existing slow trill-like CPGs to produce masculinized fast click calls.’ (p10, ln 5 – 9)

P. 11, line 6, change to "of 2 testosterone-treated female *X. laevis*"

This change has been made.

Figure 5. Some of the cross-midline connections are represented as double arrows and others as two distinct arrows. It is not clear why they are different.

In the original manuscript, the thickness of line with arrows created the illusion of double arrows, which caused confusion in depicting the projection from the PBN to the contralateral NA. In the revised manuscript, we have modified the line thickness in the revised manuscript to accurately illustrate the projections.

p. 14, line 8, and p. 27, line 24, "high-divalent (Hi-Di) saline" should be changed to high-divalent cation (Hi-Di) saline.

This change has been made.

I think that the slow and fast CPGs are more analogous to the way that the pyloric and gastric mill CPGs interact in the STG than different speeds of spinal locomotion, which can only do one thing at a time.

We appreciate this suggestion. The pyloric and gastric mill CPGs control distinct muscle groups and operate simultaneously, which is different from fast and slow CPGs in *Xenopus* that control the same muscle groups and are active at mutually exclusive timing. However, the coordination between the two CPGs is well characterized at the cellular and synaptic levels. In the Discussion section titled “Anatomically distinct organization of frequency-dependent CPGs, and their temporal coordination”, we have added a paragraph that speculates on how the timing of fast and slow trills in *Xenopus* nervous system may be coordinated using similar types of synapses the connect the pyloric and gastric mill CPGs”. (p22, ln 4 – 20)

We have kept the reference to spinal locomotion to illustrate the resemblance in the organization of the CPGs between the spinal oscillators and fast and slow CPGs of *Xenopus*. Specifically, both systems have CPGs of different rates that are made of different constituent neurons and regulate the same muscle groups, with mutually exclusive timing.

The discussion could benefit from a more complete review of what is known about comparative circuits, particularly in invertebrates, where the CPGs are well established.

We have now included a brief review of divergence of CPGs in invertebrates to provide better background for the present work in the new Discussion section titled “Evolution of *Xenopus* fast click central pattern generators”. (p24, ln 8 – p25, ln 12).